
**Daily CO2 emission for China's provinces in 2019 and 2020**

4    Duo Cui[1], Zhu Liu[1]*, CunCun Duan[2], Zhu Deng[1], Xiangzheng Deng[3], Xuanren Song[4], Xinyu Dou[1], Taochun Sun[1]

1 Department of Earth System Science, Tsinghua University, Beijing, China.

8    2 State Key Joint Laboratory of Environmental Simulation and Pollution Control, School of Environment, Beijing Normal University, Beijing, China

3 Key Laboratory of Land Surface Pattern and Simulation, Institute of Geographical Sciences and Natural Resources Research, Chinese Academy of Sciences, Beijing, China

12    4 Department of Information Systems and Business Analytics, Deakin University, Melbourne, Australia

* Corresponding authors: zhuliu@tsinghua.edu.cn

**Abstract**

Tracking China's national and regional $CO_2$ emission trends is becoming ever more crucial. The country recently pledged to achieve ambitious emissions reduction targets, however, high-resolution datasets for provincial level $CO_2$ emissions in China are still lacking. This

24    study provides daily $CO_2$ emission datasets for China's 31 provinces, including for the first time, the province of Tibet. The inventory covers the emissions from three industrial sectors (power, industry and ground transport) during 2019 to 2020, with its temporal resolution at a daily level. In addition, the variations in CO2 emissions for seasonal, weekly and holiday

periods have been uncovered at a provincial level for the first time. This new data was added to further analyze the impact that weekends and holidays have on China's $CO_2$ emissions. Over weekend periods, carbon emissions are shown to reduce by around 3%. Spring Festival meanwhile, has the greatest impact on the reduction of China's $CO_2$ emissions. This detailed

and time-related inventory will facilitate a more local and adaptive management of China's $CO_2$ emissions during both the COVID-19 pandemic's recovery and the ongoing energy transition. The data are archived at https://doi.org/10.5281/zenodo.4730175 (Cui et al., 2021).

## 1.    Introduction

China has the largest $CO_2$ emission worldwide, accounting for about 28% of global $CO_2$

emissions in 2019 (Friedlingstein et al., 2020). China is also the fastest growing country in



terms of $CO_2$ emissions (IEA, 2020). China's energy consumption and $CO_2$ emissions have been relatively stable since 2013 (Friedlingstein et al., 2019; Liu et al., 2015; Guan et al., 2018). However, since 2018, China's energy consumption and $CO_2$ emissions have shown a

new upward trend. The annual growth rates of China's Carbon emissions have exceeded 2% between 2018 and 2019 (Friedlingstein et al., 2019; Friedlingstein et al., 2020). In contrast, the growth rate of global carbon emissions is only 0.1%. China, therefore, faces considerable pressure from the international community to reduce emissions. The estimates of China's $CO_2$

emissions carry significant uncertainties, and the differences in estimates of China's carbon emissions between inventories from EDGAR, CDIAC, and CEADs approach 15% (Liu et al., 2015). Having timely and accurate $CO_2$ emission estimates based on fossil fuel combustion and cement production is therefore fundamental prerequisite to designing evidenced-based

policies for reducing China's $CO_2$ emissions.

The achievement of China's national $CO_2$ mitigation target will rely on the implementations of certain actions and policies at provincial levels. Previous studies have compiled the provincial emission inventories, which generally include the annual carbon emissions from

energy- and industry- related sectors in 30 provinces (except for Tibet) in mainland China (Shan et al., 2018). However, these studies are based on provincial energy statistics and often have a time lag of one or more years.

The timely updating of $CO_2$ emissions data is a critical step for China's provinces to achieve

the carbon neutrality plans. Moreover, the outbreak of the COVID-19 pandemic has resulted in extra uncertainties regarding China's future $CO_2$ emissions trajectory (Liu et al., 2020a; Liu et al., 2020b). During the outbreak of the COVID-19 pandemic (January-June 2020) in China, the national $CO_2$ emissions were reduced by 3.7% (-187.2 Mt $CO_2$) compared to the same

period in 2019. Therefore, there is an urgent need to obtain a timely $CO_2$ emission dataset at provincial level (Liu et al., 2020c) so that the post-COVID emission dynamics can be tracked accordingly.

In a recent study, Liu et al. (2020a; 2020b) described the *Carbon Monitor* of Fossil Fuel $CO_2$ Emissions dataset, which provides daily $CO_2$ emissions data up till December 31st, 2020, for 6 sectors and 12 largest emitting countries, plus the rest of the world as an aggregate. The global fossil fuel $CO_2$ emissions was separated into sectors of power generation (~40%

globally), industrial production (~30%), transportation (~20%, categorized as ground, air and shipping) and residential consumption (~10%). This product is evaluated against preliminary national energy usage data for all or part of the year 2020, thereby providing a full picture of all the CO2 emission drivers, including the pandemic (seasonality, working days and holidays,

weather and the economy). By acknowledging the uncertainties more than just the inventories, such a dataset can provide more up-to-date information than official inventories (UNFCCC, 2020a, b, c) and international $CO_2$ emissions datasets (BP, 2020; Crippa et al., 2020; Friedlingstein et al., 2020 ; Gilfillan et al., 2020; IEA, 2020), which have a time lag of 6 to 16

80    months after the last month of reported emissions.



In this study, based on China's daily emissions at the national scale, which are taken from the *Carbon Monitor* and provincial sectoral weight factors related to CO2 emissions, we estimate mainland China's daily $CO_2$ emissions from electricity, industry and ground transport sectors in all 31 provinces. The full names of these 31 provinces and their corresponding abbreviations are shown in Table 1. $CO_2$ emissions from Tibet are included in this dataset for the first time. This detailed and timely inventory will facilitate a more local and adaptive management of $CO_2$ emissions in the process of cutting carbon emissions and achieving carbon neutrality.

## 2. Materials and Methods

This dataset accounts for the daily changes in provincial $CO_2$ emissions in mainland China for the years 2019 and 2020. Daily provincial $CO_2$ emissions are estimated from three sectors: power, industry, and ground transport. It considers $CO_2$ emissions estimates based on administrative territories, while the emissions from international aviation and shipping are excluded (Shan et al., 2017). The estimates of national $CO_2$ emissions from 2019 to 2020 is derived from the *Carbon Monitor* dataset (data available at https://carbonmonitor.org).

### 2.1 National daily CO₂ emissions in 2019 and 2020

China's daily $CO_2$ emissions estimates are based on a near-real-time daily dataset of the global $CO_2$ emissions from fossil fuel and cement production since January 1, 2019, as published by the *Carbon Monitor* (data available at https://carbonmonitor.org/) (Liu, Ciais et al. 2020). Emission estimates from the Carbon Monitor are calculated on a national basis and by sectors, thereby gaining from past experiences in constructing annual inventories and newly compiled activity data (Liu, Ciais et al. 2020).

The *Carbon Monitor* has calculated China's daily $CO_2$ emissions since January 1, 2019. It is separated into several key emission sectors: power sector, industrial production, residential consumption, ground transport, air transport and ship transport. For the first time then, China's daily emission estimates are produced for these sectors based on regularly updated dynamic activity data.

China's near real-time activity data includes daily data for electricity generation, and monthly production data for cement, steel and other energy-intensive industrial sectors. In addition, it includes hourly traffic congestion data for 22 cities, daily maritime and aircraft transportation activity data, and previous-year fuel usage data for both residential and commercial buildings that has been corrected for air temperature.

### 2.2 Provincial distribution of daily CO₂ emissions



Daily provincial $CO_2$ emissions are estimated by daily national $CO_2$ emissions multiplied by the provincial weight factor. The sectoral provincial daily $CO_2$ emission can be obtained as follows:

$$\sum_{i,p} E_{pro} = E_{i,China} \times R_{i,p}$$

(1)

where $E_{pro}$ represents daily provincial emissions, $E_{i,china}$ refers to daily national $CO_2$ emissions, and $R_{i,p}$ denotes the weight factor of province $p$ from $i$ sector.

Due to the lack of either daily or monthly provincial energy consumption data, we used alternative indicators that reflect the provincial activity data of the corresponding sector in place of the provincial activity data.

The weight factors for each province equal the amount of provincial alternative indicators divided by the national amount of alternative indicators. The equation is as follows:

$$R_{i,p} = \frac{P_{i,p}}{P_{i,n}}$$

(2)

where $P_{i,p}$ and $P_{i,n}$ represent the provincial and national amount of alternative indicators for sector $i$, respectively. For the power, industry and ground transport sectors, the alternative indicators are thermal power generation, cement production and vehicle ownership. The data on provincial thermal power generation, cement production and vehicle ownership were obtained from China's National Bureau of Statistics of China (NBSC).

**2.3 Provincial weight factor estimation**

The provincial dataset constructed in this study includes daily $CO_2$ emission data from the three main polluting sectors (power, industry and ground transport) in China, which together account for more than 90% of the total emissions. $CO_2$ emissions from residential consumption and aviation are not considered in the provincial dataset for three reasons.

Firstly, residential and aviation totally accounted for less than 9% of China's average daily CO2 emissions (Table 2). Secondly, currently there is no suitable statistical indicator that can be used as a provincial weight factor to divide residential sector emissions from the national level into the provincial level. Thirdly, it is very difficult to count aviation emissions within provincial territories.

For the power sector, we collected monthly data on thermal power generation from 31 provinces in the Chinese mainland from January 2019 to December 2020. Due to the limitations on daily thermal power generation, we assumed that the change in ratio from provincial thermal power generation to national thermal power generation is negligible when looked at on a monthly scale. We therefore used this ratio of provincial thermal generation to national generation on a monthly scale as the provincial weight factor for the daily scale of



the power sector. Provincial weight factors from the power sector during 2019 and 2020 were
shown in Table 3.

For the industry sector, $CO_2$ emissions from steel, cement, chemicals and other industries are
168 calculated on a national scale according to the data provided by the Carbon Monitor. However,
at a provincial scale, only the data on cement production was available, with other indicators
from the steel, chemical and other industries were missing. Considering the intermediate
processes, industrial cement processes have the highest proportion of emissions in the
172 industrial sector. The ratio of the provincial production of industrial cement to the national
production was taken as the provincial weight factor for this industry sector (Table 4).

For the ground transport sector, we used the ratio of provincial vehicle ownership to national
vehicle ownership for the year 2018 as the provincial weight factor for the ground transport
sector (Table 5). Due to a lack of monthly and daily data on provincial vehicle ownership
after 2018, however, we considered the fact that there have been a change of less than 2.3% in
two years in the provincial share of vehicle ownership in the country, and thus assumed the
change in the ratio of provincial vehicle ownership to national vehicle ownership to be
negligible.

**2.4 Provincial emissions over the weekdays, weekends and holidays**

In this dataset, we add two attributions – week and holiday – to provincial daily emissions.
The attribution of "week" includes seven values: Monday, Tuesday, Wednesday, Thursday,
Friday, Saturday and Sunday. The attribution of "holiday" includes: New Year, Spring
Festival, Qingming Festival, Labour Day, Duanwu Festival, Mid-Autumn Festival, and
National Day. Among them, New Year, Qingming, Duanwu Festival and Mid-Autumn
Festival are generally 3-day holidays, Labour Day is a 5-day holiday, and Spring Festival and
November are 7-day holidays. It is worth noted that Labour Day in 2019 was four days long,
from 1st May to 4th May, and in 2020, due to COVID-19, Spring Festival was extended to 10
days (24th January to 2nd February). Since the Mid-Autumn Festival holiday overlapped with
196 the National Day holiday, in 2020 the Mid-Autumn Festival and National Day together were
an 8-day holiday (1st October to 8th October). Table 6 shows the date and duration of these
holidays in 2019 and 2020.

We calculated the average daily emissions over the weekdays, weekends and holidays from
power, industry, and ground transport in 31 provinces. For weekdays, we averaged the daily
emissions from Monday to Friday; for weekends, we averaged the daily emissions from
Saturday to Sunday; and for holidays, we calculated the average daily emissions for New
Year, Spring Festival, Qingming, Labour Day, Dragon Boat Festival and Mid-Autumn
Festival, respectively.
The formula for calculating the reduced emissions over the weekends is as follows:

$$E_{rowe} = E_{ad\_Mo2Fr} - E_{ad\_Sa2Su}$$





where $E_{rowe}$ represents reduced emissions over the weekend, $E_{ad\_Mo2Frr}$ and $E_{ad\_Sa2Su}$ refer to the average daily emissions from Monday to Friday, and average daily emissions from Saturday to Sunday, respectively.

**3. Results**

**3.1 Trends in provincial daily CO2 emissions**

*Trends in total emissions*

For most provinces, there is a valley shape in average daily emission in February (Table 7). For the year 2019, BJ province has its minimum monthly average emissions in April. The
minimum monthly average emissions of 5 provinces (SH, FJ, CQ, GX and SC) occur between May to July. YN province has its minimum monthly average emission in September. The minimum monthly average emissions for TJ, HEN and HUB provinces meanwhile, occur in October. Tibet province has its minimum average monthly emission in December. In 2020
however, the minimum monthly average emissions for all provinces was in February, except for BJ (April), SH (October) and HUB (March).
The highest average daily emissions are mainly found in summer (June to August) or/and winter (November to January), and these are called summer peak and winter peak,
respectively (Fig. 1). The provinces that follow this pattern for the summer peak emissions are HEB, SX, IM, LN, JL, HLJ, HEN, Tibet, QH, NX and XJ. The remaining provinces, including BJ, TJ, SH, JS, ZJ, AH, FJ, JX, SD, HUB, HUN, GD, GX, HAN, CQ, SC, GZ, YN, SAX and GS, follow the winter peak emissions pattern (Table 7).

Referring to minimum daily emissions (Table 8), this day fell on the 5th February for 25 provinces (except for BJ, SH, HEN, CQ, SC and Tibet), which was the first day of the seven-day holiday for the Spring Festival in 2019. From those that did not follow this trend,
BJ and SH province emitted their minimum value of $CO_2$ on the 1st May, which is the first day of the four-day holiday of Labour Day in 2019; for HEN, CQ and SC it was on the 1st October in 2019, the first day of the seven-day holiday of National Day; and for Tibet it was on the 1st December in 2019. In 2020, 14 provinces, including TJ, HEB, LN, JL, HLJ, JS, ZJ,
SD, HEN, GD, QH and Tibet emitted their minimum amount of emissions on the 9th February; 9 provinces, including SX, IM, AH, FJ, JX, SAX, GS, NX, AND XJ, had their minimum daily emission day on the 13th February; 8 provinces (SH, HUN, GX, HAN, CQ, SC, GZ, YN), had it on the 2nd October, which was the second day of the joint Mid-Autumn Festival
and National Day holidays; for BJ province it was on the 4th April; and for HUB province, it was the 1st March.

Referring to maximum daily emissions, in 2019, the daily emissions in 20 provinces,
including BJ, TJ, SH, JS, ZJ, AH, JX, SD, HUB, HUN, GD, GX, HAN, CQ, SC, GZ, YN, SAX, GS, QH, and XJ, reached their maximum value on a day between November to January (Table 8); for 8 provinces, including HEB, SX, IM, LN, HEN, Tibet and NX were on the 26th



July; and HLJ, FJ and QH were all on the 9th September. In 2020, maximum daily emissions occurred on a day between November and January in all provinces except for IM, JL, FJ, HEN, Tibet, HLJ and QH provinces. IM, JL, FJ, HEN and Tibet provinces emitted their maximum $CO_2$ on a day between June and August, and HLJ and QH emitted their maximum $CO_2$ on the 22nd October.

### *Trends in emissions from the power sector*

Provincial daily $CO_2$ emissions from mainland China's power sector show two trends during an annual period: the "W" shape trend and the "U" shape trend. In most Chinese provinces (25 provinces, except for the six provinces of Tibet, QH, SAX, GS, SC and GZ), daily $CO_2$ emissions show two peaks and two valleys during an annual period, which we called the "W" trend (Fig. 2, a-aa). The two peaks occur in July-August and December-January, and we refer to them as the summer peak and the winter peak, respectively. The summer peak in the power sector is due to the widespread use of air conditioning, whereas the winter peak is due to heating. The valleys occur during the Spring Festival period and the National Day holiday, which are the two most important seven-day holidays in China.

The remaining six provinces (Tibet, QH, SAX, GS, SC and GZ), which are all western provinces, daily $CO_2$ emissions show the pattern of one peak and one valley during an annual period, which we call the "U" trend (Fig. 2, ab-ae). These western provinces have lower average summer temperatures and therefore have no summer peak of emissions. However, the winter peaks can be seen as daily $CO_2$ emissions from the power sector in these provinces.

### *Trends in emissions from industry sector*

Provincial daily $CO_2$ emissions from the industry sector in mainland China show two trends during an annual period: the "inverted-U" shape trend and the "line" shape trend.
In 16 northern provinces of China, including LN, HLJ, JL, HEN, IM, GS, SX, SD, HEB, SAX, NX, XJ, QH, Tibet, TJ, and BJ, daily $CO_2$ emissions remain high from June to November, and then drop down from December to January, which we called the "inverted-U" shape trend (Fig. 3).
In 15 southern provinces of China, including YN, JX, FJ, GX, JS, HUB, GZ, GD, HAN, SH, CQ, AH, ZJ, SC, and HUN, daily $CO_2$ emissions reach their lowest point during the Spring Festival holiday, then slowly rise and reach the peak at the end of the year, which we call the "line" shape trend (Fig. 4).

### *Trends in emissions from ground transport*

The impact that holidays have on daily $CO_2$ emissions trends is more pronounced in the ground transportation sector (Fig. 6). For the year 2020 however, COVID-19 has had a more



lasting impact on the $CO_2$ emissions from the transportation sector than on those from the power and industry sectors, which is consistent with previous studies (Liu et al., 2020a; Liu et al., 2020b; Liu et al., 2020c). $CO_2$ emissions from the ground transportation sector reached the same level until September in 2020 for the same period in 2019.

***Trends in sectoral contribution***

Fig. 6 shows the trends in the contribution rate of emissions from power, industry and ground transportation. In some provinces, such as in SD, JS, GD, ZJ, AH, SC, HUB, GX, GZ, FJ, 304 HUN, JX, YN, CQ and HAN, the contribution of sectoral emissions varies little within one year. However, in HEB, IM, HEN, SX, XJ, SAX, LN, NX, JL, HLJ, GS, SH, TJ, BJ, QH and Tibet, the contribution of sectoral emissions shows a large variation within one year. Among the provinces that show large changes in sectoral emission contribution trends, except for SH, 308 BJ and Tibet, all the emissions-contributing trends show that the largest contribution from the power sector is during winter, there is an increased share from the industry sector in summer, and the contribution from the transportation sector is relatively stale year-round.

In GS and QH provinces, the power sector accounts for more than 70% of the total daily emissions in winter and less than 30% of the total daily emissions in summer, while the industry sector accounts for more than 60% of the total daily emissions in winter. In the provinces of HLJ, JL, LN, SAX and HEB, the power sector contributes around 60% of the 316 total daily emissions in winter, while in summer, the contribution from the industry sector reaches over 40%, which is close to or exceeds the contribution of daily emissions from the power sector. In SH and BJ, the contribution to total daily emissions from the power sector reaches its maximum in winter, and its minimum in the periods of April-May and 320 September-October. The ground transportation sector is the sector that contributes the second largest volume of emissions in the period of April-May and September-October. This is especially true in BJ, where the transportation sector accounts for about 40% of total emissions in April-May and September-October, which is close to the emissions from the 324 power sector during that period. Meanwhile, $CO_2$ emissions in Tibet are mostly contributed by the industry sector. In summer, the industry sector contributes over 90% of the daily $CO_2$ emissions, while in winter, it decreases to around 80% with the rest of the daily emissions mainly coming from ground transport.

**3.4 The effect of the weekend on CO2 emissions**

In a week, the lowest average daily emissions are observed on Saturday and Sunday (Fig. 7). 332 Emission reductions over the weekend were seen to exceed 50 thousand tons of $CO_2$ per day in the provinces of SD, GD and JS (Fig. 8a). Among 28 of China's 31 provinces, a substantial decrease in $CO_2$ emissions can be seen over the weekend: the 17 provinces of HEB, HEN, ZJ, IM, AH, SX, SC, XJ, FJ, GX, HUB, HUN, SAX, JX, LN, YN and GZ, have a weekend 336 emissions reduction that ranges from 20-40 thousand tons $CO_2$ (Fig. 8a), while the remaining 11 provinces of CQ, NX, GS, HLJ, JL, SH, TJ, BJ, HAN, QH and Tibet, show a weekend emissions reduction of less than 12 thousand tons of $CO_2$ (Fig. 8a).



Out of the 31 provinces, BJ has the most prominent reduction of emissions on weekends, accounting for 4.03% of the average daily emissions on weekdays. GZ meanwhile, has the lowest reduction of emissions on weekends, accounting for only 2.35% of average daily $CO_2$ emissions on weekdays. In the remaining provinces, the reduction of emissions over the

weekend is equivalent to around 3% of average daily $CO_2$ emissions on weekdays (Fig. 8b).

The decreased emissions from the power, industry and ground transport sectors on weekends vary across provinces. However, for the provinces of XJ, SX, NX, SD, JS, AH, TJ, GS, SAX,

and IM, we see that the main reduction in emissions comes from the power sector (Table 9), because the power sector is also the main contributor to those emissions generally (Fig. 6). For the provinces of Tibet, HUB, JX, HUN, YN, GZ, FJ, GX, HAN, CQ and SC, the industrial sector is also both the main driver of $CO_2$ emissions and the sector with the highest

reductions on weekends. For China's northeast region, GD, ZJ, HEN, HEB, as well as for BJ and SH, the reduction of $CO_2$ emissions on weekends mainly comes down to ground contribution sector, which, especially in BJ and SH, is responsible for over 70% and nearly 50% of the emissions reduction on weekends, respectively.

## 3.5 The effect of holidays on $CO_2$ emissions

China has seven major holidays, including New Year, Spring Festival, Qingming Festival, Labour Day, Duanwu Festival, Mid-Autumn Festival and National Day. The dates and

durations of the holidays we mentioned above in 2019 and 2020 are shown in Table 6. During the period of 2019-2020, the average daily emissions on holidays (dark blue lines in Fig. 9) can be seen to be less than average daily emissions on normal days (light blue lines in Fig. 9), which means that $CO_2$ emissions are reduced on holidays.

In 2019, compared to other holidays, the largest reduction in average daily $CO_2$ emissions was recorded during Spring Festival (Fig. 10a). The maximum reduction of average daily $CO_2$ emissions in that period reached 375.08 thousand tons of $CO_2$ in JS province, while the

minimum reduction of average daily $CO_2$ emissions in that period reached 5.5 thousand tons of $CO_2$ in Tibet. Considering the durations of China's national holidays (Table 6), the cumulative reduction of $CO_2$ emissions on Spring Festival are the largest compared to other holidays (Fig. 10c). During the 2019 Spring Festival, the cumulative reduction of $CO_2$

emissions reached 30075.89 thousand tons, which is equivalent to 0.33% of the total national emissions in 2019. The cumulative holidays in China during the year 2019 resulted in a total reduction of 67450.21 thousand tons of $CO_2$, which represents 0.75% of China's annual emissions. Between them, the provinces of JS and SD had a total reduction of more than 5000

thousand tons of $CO_2$ during these holidays in 2019, thereby contributing 15.87% of the total reductions in that period.



In 2020, the joint Mid-Autumn Festival and National Day holidays led to an 8-day holiday whose impacts on $CO_2$ emissions exceeded that of the Spring Festival (Fig. 10b). The possible reason for this is that $CO_2$ emissions on normal days had been reduced due to the COVID-19 pandemic, and thus 2020's Spring Festival, which fell during the COVID-19 pandemic, had a less impressive reduction rate. However, by October 2020, China's daily

emissions had already recovered from the COVID-19 pandemic and exceeded the level for the same period of 2019. Thus, compared to the Spring Festival period, average daily emissions were able to show a greater reduction compared to normal days when reaching the joint holidays of Mid-Autumn and National Day. Average daily emissions did show their

second largest reduction during Spring Festival however. The impact of the other four holidays on average daily emissions meanwhile, these being New Year, Qingming Festival, Labour Day and Duanwu Festival, are less than the combined reduction from the joint Mid-Autumn Festival and National Day, and Spring Festival.

During the joint holidays of the Mid-Autumn Festival and National Day, the reduction of average daily emissions in JS, SD and GD provinces exceeded 400 thousand tons of $CO_2$ (Fig. 11b); for the 18 provinces of AH, ZJ, HN, IM, HUB, HEB, SC, GZ, SX, FJ, SAX, HUN, XJ,

YN, JX, CQ and LN, it ranged from 100 to 310 thousand tons of $CO_2$; and for the remaining provinces of NX, GS, SH, HLJ, TJ, JL, BJ, HAN, QH and Tibet, it was between 18 and 92 thousand tons of $CO_2$. During the period of the joint holidays of Mid-Autumn Festival and National Day (8 days, Table 6), the cumulative reduction of CO2 emissions at a national level

reached 44447.45 thousand tons of CO2, which is close to 0.5% of the national emissions for 2020. The total reduction of CO2 emissions over the 7 holidays was 91350.96 thousand tons of CO2, which is equivalent to 1.01% of national annual emissions. The provinces of SD, JS and GD contribute the highest reduction of emissions during holiday periods, accounting for

8.90%, 7.43% and 7.53% respectively, of China's total reduction of emissions on holidays.

## 4. Comparison with CEADs dataset

Although the time range of the CEADs (Carbon Emission Accounts & Datasets, https://www.CEADs.net.cn/ ) dataset and this study do not match up, the proportion of provincial emissions changed little in those two years. The most recent available data in the CEADs inventory is for the year 2018, and thus this was used for comparison with the

sectoral provincial data in this study for the year 2019. In addition, our inventory estimates $CO_2$ emissions in Tibet, while the CEADs dataset does not include them. In order to carry out a comparison with the CEADs's provincial contributions, this inventory only considers the provincial contributions from 30 provinces (which excludes Tibet). Fig. 11 shows the

provincial contribution differences between the inventories from the CEADs and those from this study for the power, industry and ground transportation sectors.

When studying the power sector, the provincial emission contributions shown in the CEADs

dataset and those in this study show similar values. For the power sector, the provincial emissions in this study (Fig. 11d) are also consistent with those in the CEADs (Figs. 11a and



11d). The top 10 $CO_2$ emission contributors are the provinces of SD, IM, JS, GD, SX, HEB, XJ, HEN, ZJ and AH, both in the CEADs dataset and in this inventory. For the industry sector (Figs.11b and 11e), a large difference in the emissions contribution rate can be seen for HEB province between the CEADs dataset and this study. The CEADs inventory states that HEB province accounts for nearly 15% of the national $CO_2$ emissions coming from the power sector, while our inventory considers it to be only 4.34%. HEB province is the largest emissions emitter in the CEADs dataset, while our inventory ranks it as the 13th. This difference may be due to the fact that emissions from cement production processes are not the main emissions driver in HEB's industry sector. Regarding ground transportation, according to the CEADs dataset, SH contributes around 7%, while this inventory states that it only contributes 1.70%. However, the contribution ratio for the provinces of SD, JS, ZJ, HEN and HEB, were higher in this inventory than in CEADs's estimates.

The likeliest cause for the discrepancies of sectoral provincial contributions between the CEADs dataset and this study, is the method used in each inventory. The CEADs calculates provincial emissions based on energy consumption during sectoral processes. However, it is difficult to estimate daily emissions based on energy-related methods due to a lack of daily energy consumption data. To calculate the provincial weight factor, estimate the daily provincial weight factors and improve the time precision of the emission inventory to a daily level, this inventory uses high temporal resolution alternative indicators, such as thermal power generation for the power sector, cement production for the industry sector and vehicle ownership for ground transportation. However, some uncertainties will still be introduced due to the poor consistencies between alternative indicators and actual emission processes in some provinces.

## 5. Source of uncertainty

The uncertainties in this inventory come from two sources: the uncertainties of near-real national $CO_2$ emissions from the *Carbon Monitor*, and the uncertainties from the estimates of each provincial weight factor. The uncertainties of near-real national $CO_2$ emissions from the Carbon Monitor have been discussed in detail in (Liu et al, 2020a, b). The uncertainties from the emissions from the power and industry sectors come from monthly statistical data of thermal power generation and cement production. Uncertainties from the monthly statistics were derived from 10,000 Monte Carlo simulations carried out to estimate a 68% confidence interval (1 sigma) for provincial thermal power generation and cement production in China. The uncertainties of the ground transportation sector mainly come from the inter-annual variability of provincial vehicle ownership, which are based on the estimates in the annual data of vehicle ownership from the China Bureau of Statistics within (±2.3%).

## 6. Conclusions

Estimating China's provincial $CO_2$ emissions is fraught with problems, such as data availability and the time lag of one or more years in the data itself. In the context of a



sustained COVID-19 pandemic and China's commitment to achieve its peak carbon emissions before 2030 and then drop to carbon neutrality before 2060, annual provincial emission inventories have become untenable. The provincial daily $CO_2$ emissions dataset presented here increases the temporal resolution of the emissions inventory and estimates the daily $CO_2$ emissions from the power, industry and ground transportation sectors in 31 provinces of mainland China. This study also notably includes the $CO_2$ emission estimates for

Tibet for the first time; something which was not done previously due to the lack of available energy-related statistic data. This dataset adds the two attributes of the "week" and "holiday" to provincial daily emission, which can be used to analyse the impact of weekends and holidays on China's $CO_2$ emissions.

The provincial emissions based on the estimates in this inventory are in good agreement with those in the *CEADs* dataset. However, this inventory improves the temporal resolution to a daily level compared to only annual emissions estimates provided in the *CEADs* dataset. This dataset will be near-real time updated (may be one month behind the actual time). However,

more work is still required in order to improve the provincial daily CO2 emission estimates from the lower emitting sectors, such as the residential, aviation and shipping sectors.

## 7. Data available

The national daily $CO_2$ emissions from Carbon Monitor, provincial weight factor from China Bureau of Statistics, provincial daily CO2 emissions in 31 provinces of mainland China from 2019 to 2020, and Yearly comparison results of CEADs and This inventory can be accessed

at https://doi.org/10.5281/zenodo.4730175 (Cui et al., 2021).

## Competing interests

The author declares that they have no conflict of interest.

## Acknowledgements

Cuncun Duan of Beijing Normal University assisted with the calculation of provincial weight
factor. Zhu Deng of THU provided provincial monthly data of thermal power generation, cement production and vehicle ownership. The assistance of Xinyu Dou in calculating the uncertainty of provincial emissions from ground transport sector. This work builds on the legacy of Carbon Monitor, especially the work of Zhu Liu, Philippe Ciais and Zhu Deng.

## Financial support

This research has been supported by the funding from the National Natural Science
Foundation of China (grant 71874097 and 41921005), Beijing Natural Science Foundation (JQ19032).

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

NX, XJ, CQ, BJ, TJ, SH, HAN, Tibet, QH, SAX, GS, SC and GZ are abbreviations for the
names of these 31 provinces. The full names corresponding to these abbreviations are shown
in Table 1.

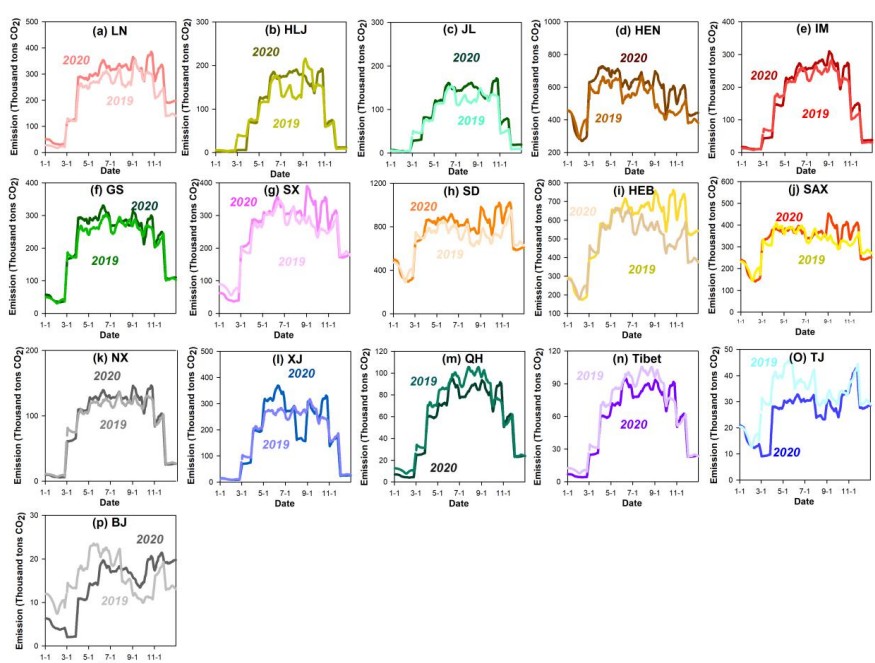

**Figure 3.** Daily $CO_2$ emissions from the industry sector in 16 of China's provinces from 2019 to 2020. LN, HLJ, JL, HEN, IM, GS, SX, SD, HEB, SAX, NX, XJ, QH, Tibet, TJ, and BJ are abbreviations of the names of 16 provinces in northern China. The full names corresponding to these abbreviations are shown in Table 1.

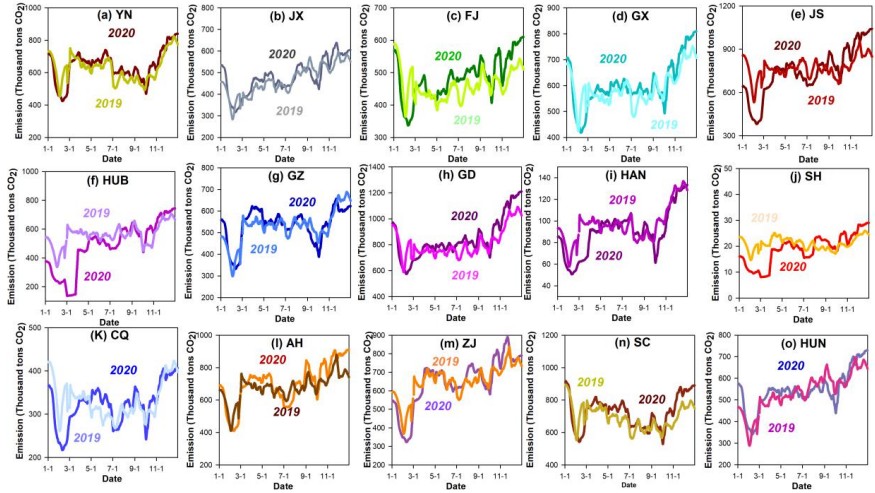

**Figure 4.** Daily $CO_2$ emissions from the industry sector in 15 of China's provinces from 2019 to 2020. YN, JX, FJ, GX, JS, HUB, GZ, GD, HAN, SH, CQ, AH, ZJ, SC, and HUN are



abbreviations of the names of these 15 provinces in southern China. The full names corresponding to these abbreviations are shown in Table 1.

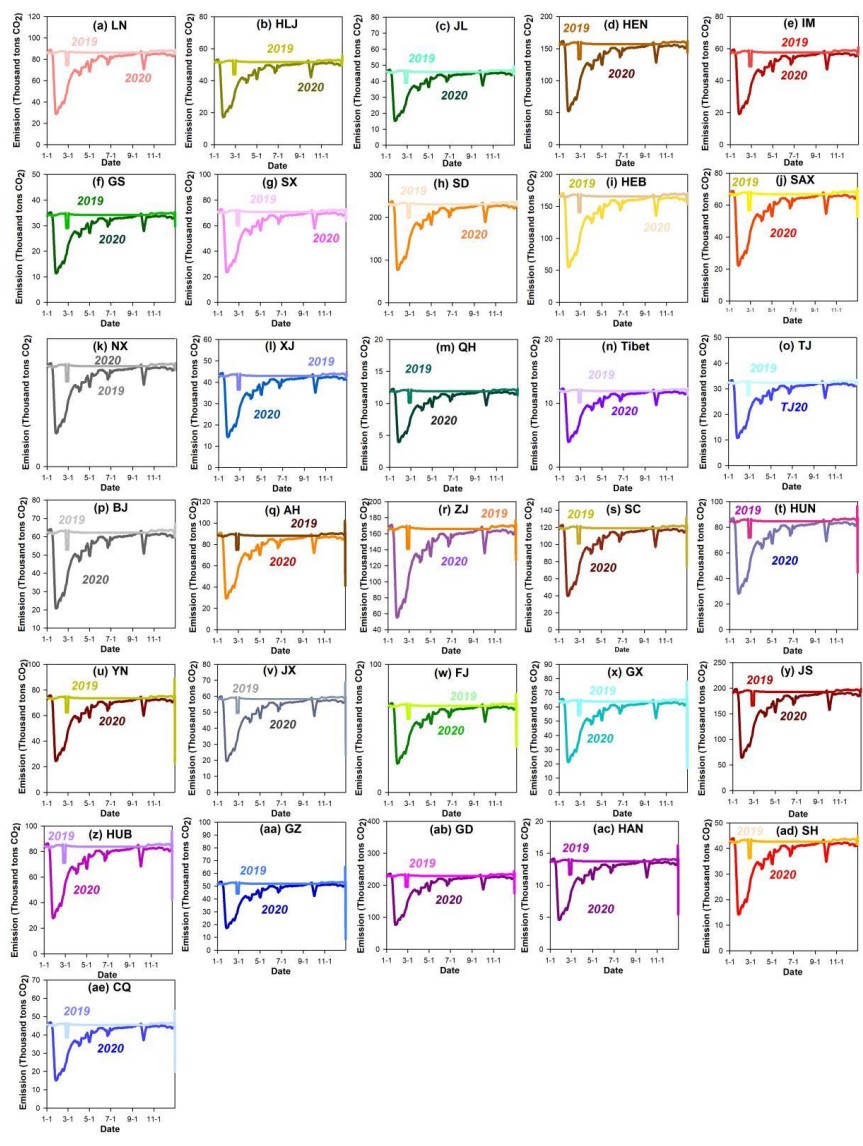

**Figure 5.** Daily CO$_2$ emissions from the ground transportation sector in China's 31 provinces from 2019 to 2020. YN, JX, FJ, GX, JS, HUB, GZ, GD, HAN, SH, CQ, AH, ZJ, SC, and HUN are abbreviations of the names of these 31 provinces in southern China. The full names corresponding to these abbreviations are shown in Table S1.



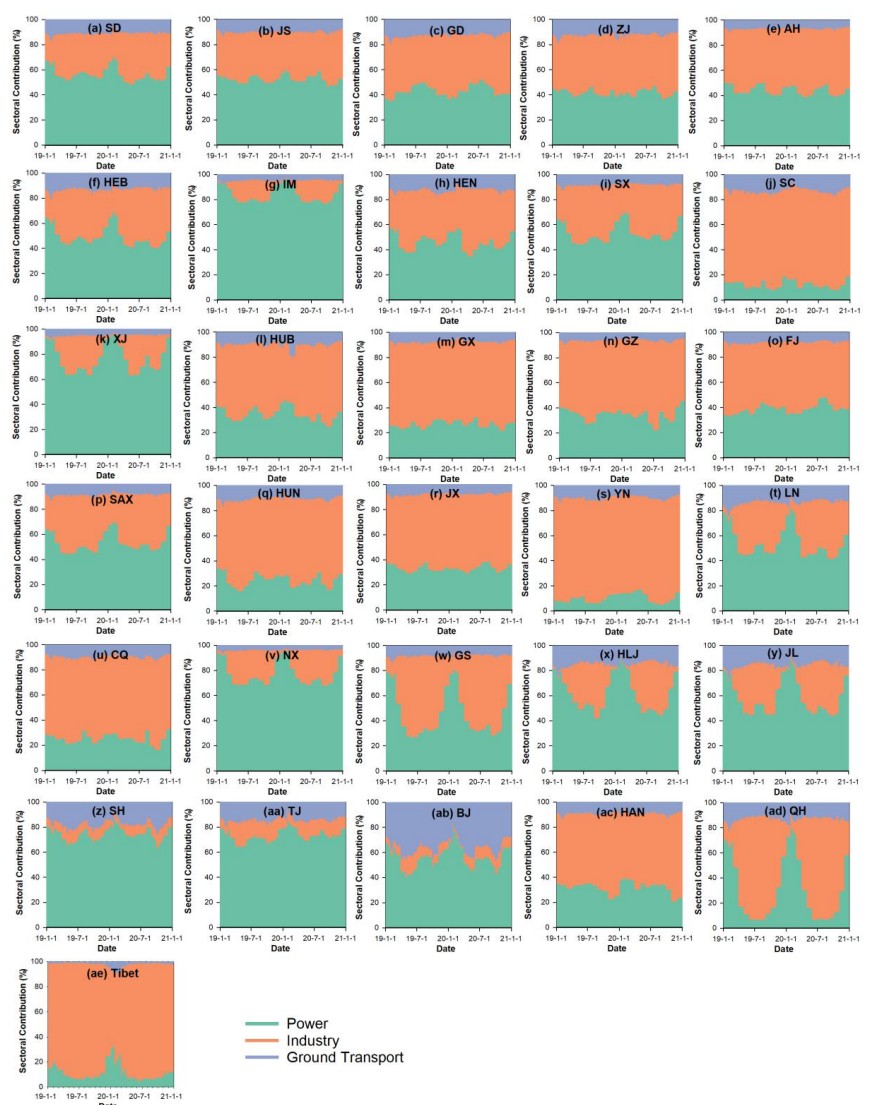

**Figure 6**. sectoral contribution to provincial daily $CO_2$ emissions in 31 provinces in China.
Green, orange and blue refer to the power, industry and ground transport sectors, respectively.
SD, JS, GD, ZJ, AH, HEB, IM, HEN, SX, SC, XJ, HUB, GX, GZ, FJ, SAX, HUN, JX, YN,
LN, CQ, NX, GS, HLJ, JL, SH, TJ, BJ, HAN, QH and Tibet are abbreviations for the names
of these 31 provinces. The full names corresponding to these abbreviations are shown in
Table 1.

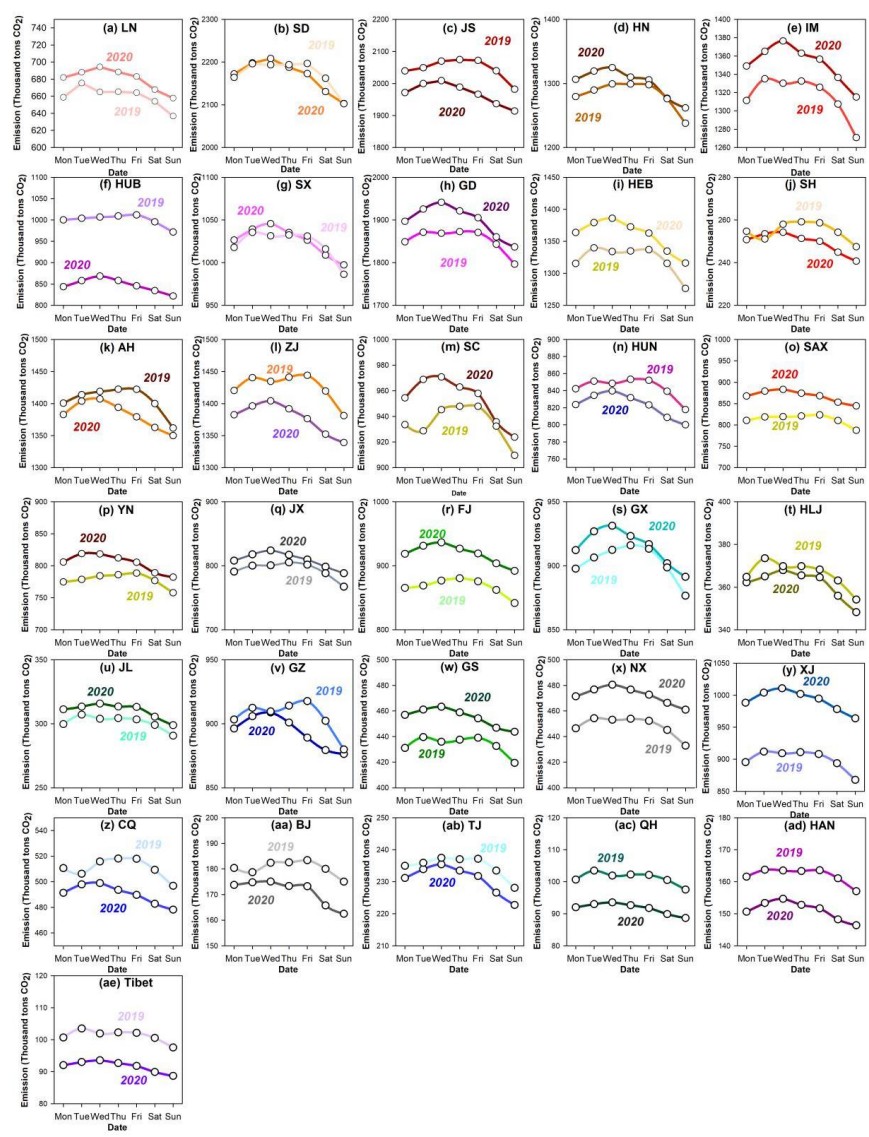

**Figure 7. Average daily CO₂ emissions from Monday to Sunday.** SD, JS, GD, ZJ, AH,
HEB, IM, HEN, SX, SC, XJ, HUB, GX, GZ, FJ, SAX, HUN, JX, YN, LN, CQ, NX, GS, HLJ,
JL, SH, TJ, BJ, HAN, QH and Tibet are abbreviations of the names of 31 Chinese provinces.
The full names corresponding to these abbreviations are shown in Table 1.



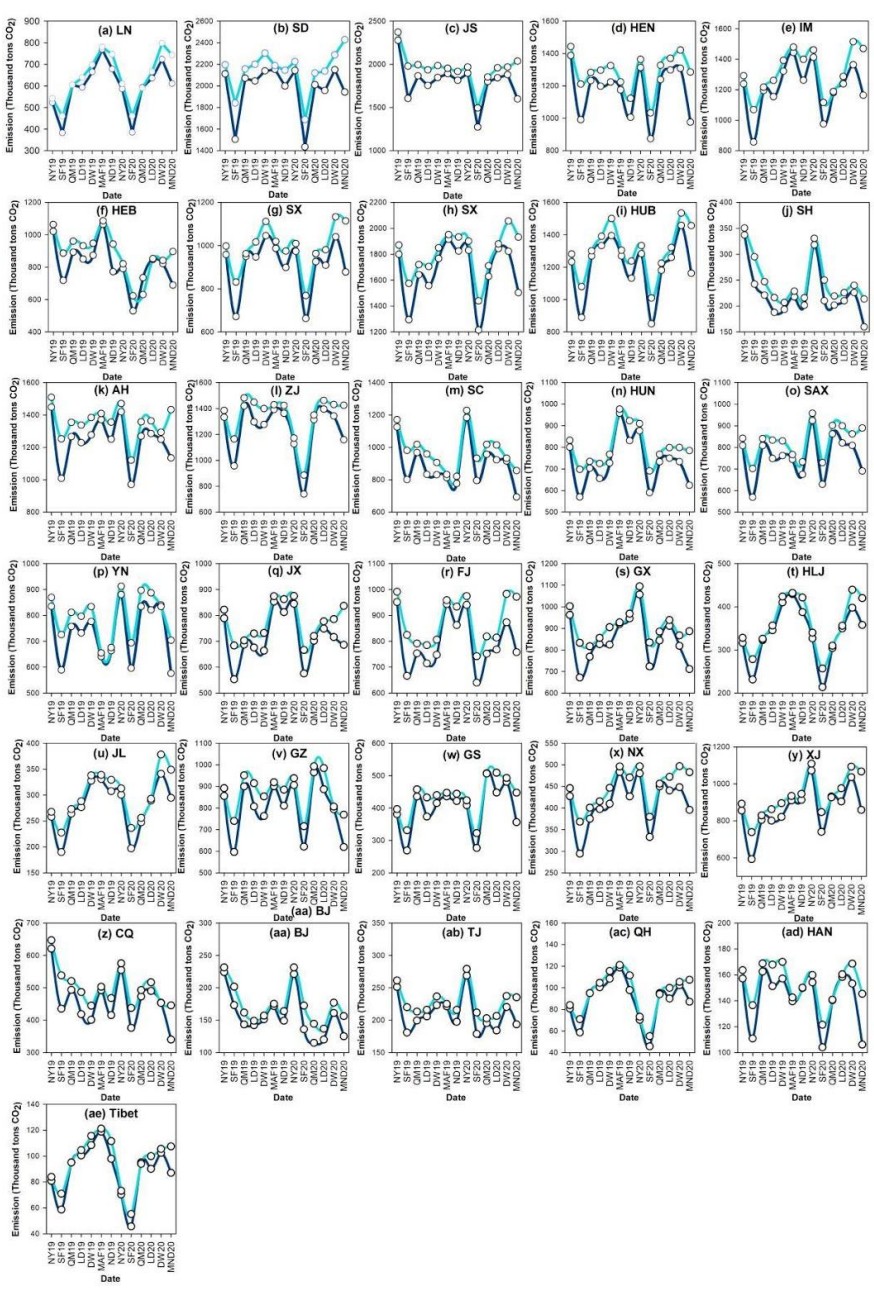

**Figure 9. Average daily CO2 emissions in 31 provinces during holiday periods.** Dark blue
lines represent the average daily emissions on holidays, while light blue lines represent the
average daily emissions for the periods of 15 days both before and after these holidays. SD,
JS, GD, ZJ, AH, HEB, IM, HEN, SX, SC, XJ, HUB, GX, GZ, FJ, SAX, HUN, JX, YN, LN,

CQ, NX, GS, HLJ, JL, SH, TJ, BJ, HAN, QH and Tibet are all abbreviations of the names of
these 31 Chinese provinces. The full names corresponding to these abbreviations are shown in
Table S1. NY, SF, QM, LD, DW, MAF, and ND represent the Chinese holidays of New Year,
Spring Festival, Qingming, Labour Day, Duanwu, Mid-autumn Festival and National Day,
respectively. 19 and 20 refer to 2019 and 2020. MND20 stands for the joining of
Mid-Autumn Festival and National Day in 2020.

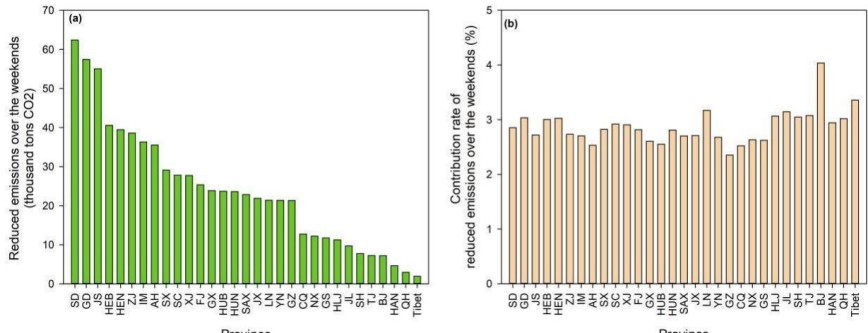

**Figure 8. Reduced emissions over the weekends (a), and contribution rate of reduced emissions over the weekends (b).** SD, JS, GD, ZJ, AH, HEB, IM, HEN, SX, SC, XJ, HUB, GX, GZ, FJ, SAX, HUN, JX, YN, LN, CQ, NX, GS, HLJ, JL, SH, TJ, BJ, HAN, QH and
Tibet are abbreviations of the names of the 31 Chinese provinces. The full names corresponding to these abbreviations are shown in Table 1.

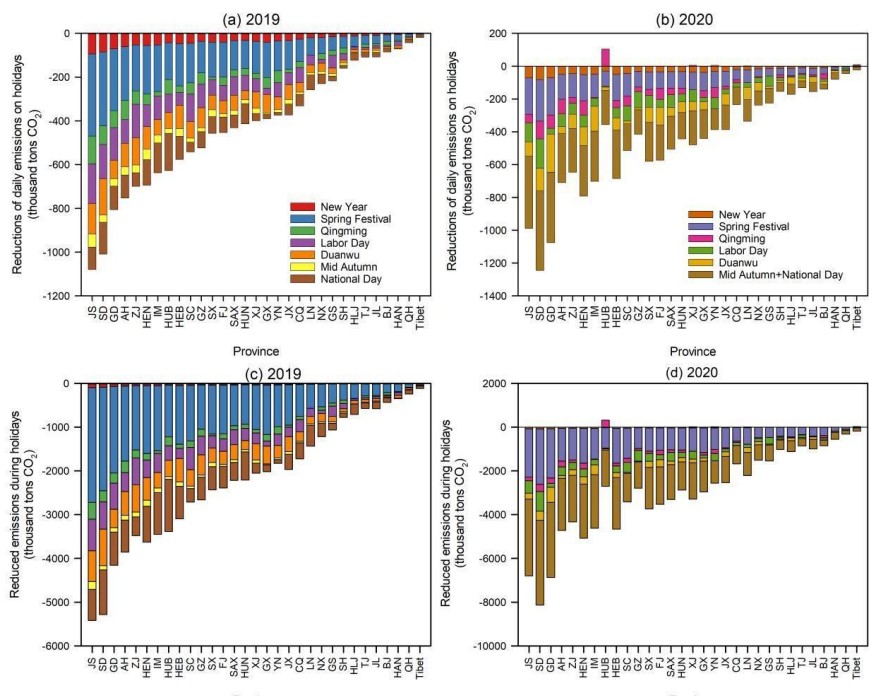

**Figure 10. Reductions of daily emissions on holidays (a and b) and reduced emissions during holidays (c and d).** SD, JS, GD, ZJ, AH, HEB, IM, HEN, SX, SC, XJ, HUB, GX, GZ, FJ, SAX, HUN, JX, YN, LN, CQ, NX, GS, HLJ, JL, SH, TJ, BJ, HAN, QH and Tibet are abbreviations of the names of the 31 Chinese provinces. The full names corresponding to these abbreviations are shown in Table S1. New Year, Spring Festival, Qingming Festival, Labour Day, Duanwu Festival, Mid-Autumn Festival and National Day refer to holidays in China.

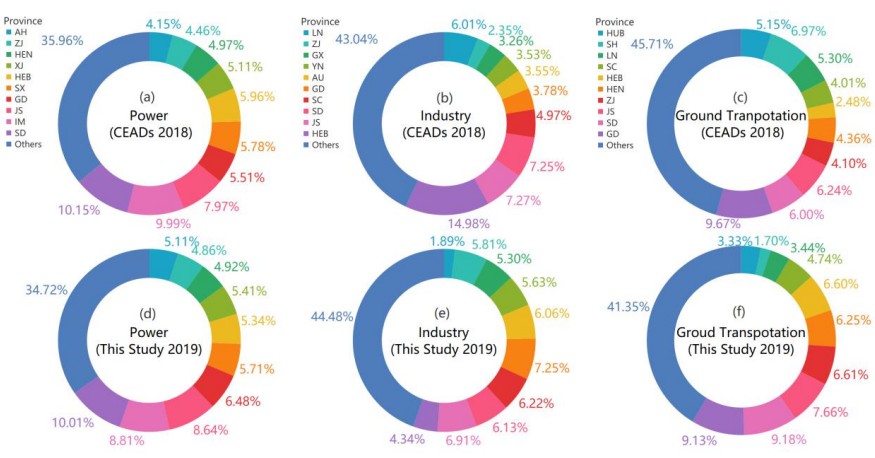



**Figure 11. Comparison of the provincial contributions to national emissions in this study to those reported by the CEADs dataset**

**Tables**

**Table 1. The abbreviation of provincial names in mainland China**

| Abbr | Full name | Abbr | Full name | Abbr | Full name |
|------|-----------|------|-----------|------|-----------|
| LN | Liaoning | IM | Inner Mongolia | HUB | Hubei |
| SD | Shandong | SH | Shanghai | NX | Ningxia |
| JS | Jiangsu | HAN | Hainan | XJ | Xinjiang |
| HN | Henan | FJ | Fujian | SX | Shanxi |
| CQ | Chongqing | GX | Guangxi | HLJ | Heilongjiang |
| BJ | Beijing | AH | Anhui | HEB | Hebei |
| TJ | Tianjin | ZJ | Zhejiang | GD | Guangdong |
| HUN | Hunan | YN | Yunnan | JL | Jilin |
| JX | Jiangxi | Tibet | Tibet | QH | Qinghai |
| SAX | Shaanxi | GS | Gansu | SC | Sichuan |
| GZ | Guizhou | | | | |

**Table 2. National average daily emissions from the residential and aviation sectors for the years 2019 and 2020.**

| Year | Total | Residential | | Aviation | |
|------|-------|-------------|--|----------|--|
| | | Emissions (Mt CO2) | contribution rate (%) | Emissions (MtCO2) | contribution rate (%) |
| 2019 | 28.66 | 0.17 | 0.60 | 2.20 | 7.69 |
| 2020 | 28.73 | 0.14 | 0.49 | 2.21 | 7.69 |

**Table 3. Provincial monthly weight factor from the power sector in 2019 (a) and 2020 (b).**

| Province | (a) 2019 (%) | | | | | | | | | | | |
|----------|-----|-----|-----|-----|-----|-----|-----|-----|-----|-----|-----|-----|
| | Jan | Feb | Mar | Apr | May | Jun | Jul | Aug | Sep | Oct | Nov | Dec |
| BJ | 1.06 | 1.06 | 0.92 | 0.60 | 0.59 | 0.63 | 0.80 | 0.78 | 0.82 | 0.71 | 0.92 | 0.90 |
| TJ | 1.40 | 1.40 | 1.33 | 1.21 | 1.34 | 1.43 | 1.46 | 1.16 | 1.32 | 1.24 | 1.38 | 1.59 |
| HEB | 5.60 | 5.60 | 5.38 | 5.33 | 5.81 | 5.96 | 5.58 | 4.95 | 4.74 | 5.03 | 5.22 | 4.99 |
| SX | 5.63 | 5.63 | 5.34 | 5.82 | 6.08 | 6.14 | 6.13 | 5.70 | 5.39 | 5.51 | 5.67 | 5.53 |
| IM | 8.20 | 8.20 | 8.51 | 9.07 | 9.10 | 9.48 | 9.38 | 8.99 | 9.42 | 9.26 | 8.66 | 7.72 |
| LN | 2.87 | 2.87 | 2.86 | 2.59 | 2.67 | 2.78 | 3.18 | 2.92 | 3.02 | 2.91 | 2.87 | 2.85 |
| JL | 1.45 | 1.45 | 1.45 | 1.37 | 1.29 | 1.34 | 1.46 | 1.39 | 1.26 | 1.29 | 1.43 | 1.48 |
| HLJ | 1.82 | 1.82 | 1.78 | 1.87 | 1.91 | 1.85 | 1.72 | 1.45 | 1.52 | 1.80 | 1.90 | 1.83 |
| SH | 1.91 | 1.91 | 1.78 | 1.45 | 1.25 | 1.20 | 1.44 | 1.70 | 1.31 | 1.29 | 1.35 | 1.68 |
| JS | 8.93 | 8.93 | 9.09 | 9.07 | 8.72 | 8.37 | 8.85 | 8.87 | 8.34 | 8.41 | 8.27 | 7.99 |



| Province | Jan | Feb | Mar | April | May | Jun | Jul | Aug | Sep | Oct | Nov | Dec |
|---|---|---|---|---|---|---|---|---|---|---|---|---|
| ZJ | 4.24 | 4.24 | 5.56 | 5.59 | 5.07 | 4.76 | 4.36 | 5.20 | 4.77 | 5.02 | 4.69 | 4.89 |
| AH | 5.15 | 5.15 | 4.94 | 5.00 | 5.24 | 5.40 | 5.49 | 5.50 | 4.88 | 4.69 | 4.74 | 5.01 |
| FJ | 2.30 | 2.30 | 2.35 | 2.61 | 2.78 | 2.40 | 2.49 | 2.98 | 3.34 | 3.27 | 2.72 | 2.75 |
| JX | 2.09 | 2.09 | 1.89 | 2.03 | 2.00 | 1.89 | 2.03 | 2.52 | 2.32 | 2.37 | 2.17 | 2.18 |
| SD | 10.06 | 10.06 | 9.92 | 10.44 | 10.70 | 10.86 | 10.17 | 9.01 | 10.10 | 10.24 | 9.92 | 9.16 |
| HEN | 5.61 | 5.61 | 4.42 | 4.40 | 4.62 | 5.28 | 6.07 | 5.33 | 4.76 | 4.07 | 4.24 | 4.44 |
| HUB | 2.97 | 2.97 | 2.71 | 2.46 | 2.55 | 2.62 | 3.02 | 3.29 | 3.22 | 2.22 | 2.73 | 3.07 |
| HUN | 1.93 | 1.93 | 1.34 | 1.24 | 1.07 | 1.33 | 1.55 | 2.26 | 2.29 | 1.97 | 1.97 | 2.06 |
| GD | 4.72 | 4.72 | 6.15 | 6.50 | 6.57 | 7.72 | 7.16 | 7.08 | 7.49 | 7.62 | 6.23 | 5.91 |
| GX | 1.76 | 1.76 | 1.64 | 1.77 | 2.42 | 1.95 | 1.33 | 1.58 | 2.34 | 2.77 | 2.45 | 1.99 |
| HAN | 0.39 | 0.39 | 0.44 | 0.51 | 0.52 | 0.50 | 0.46 | 0.36 | 0.35 | 0.41 | 0.31 | 0.34 |
| CQ | 1.25 | 1.25 | 1.11 | 1.18 | 0.91 | 0.81 | 0.80 | 1.19 | 1.11 | 0.84 | 1.08 | 1.26 |
| SC | 1.15 | 1.15 | 1.28 | 1.31 | 0.80 | 0.83 | 0.65 | 1.01 | 0.66 | 0.62 | 0.76 | 1.43 |
| GZ | 2.44 | 2.44 | 3.04 | 3.04 | 2.73 | 1.93 | 1.73 | 2.42 | 2.81 | 2.79 | 2.89 | 3.00 |
| YN | 0.49 | 0.49 | 0.51 | 0.76 | 0.78 | 0.85 | 0.35 | 0.32 | 0.35 | 0.57 | 0.85 | 0.92 |
| Tibet | 0.00 | 0.00 | 0.00 | 0.01 | 0.01 | 0.01 | 0.01 | 0.01 | 0.01 | 0.01 | 0.01 | 0.01 |
| SAX | 3.66 | 3.66 | 3.69 | 3.45 | 3.37 | 3.16 | 3.11 | 3.08 | 2.99 | 2.87 | 3.88 | 3.97 |
| GS | 2.10 | 2.10 | 2.07 | 1.46 | 1.05 | 1.03 | 1.00 | 1.18 | 1.18 | 1.34 | 1.82 | 2.01 |
| QH | 0.40 | 0.40 | 0.35 | 0.15 | 0.11 | 0.07 | 0.06 | 0.06 | 0.12 | 0.15 | 0.26 | 0.33 |
| NX | 2.82 | 2.81 | 2.67 | 2.42 | 2.76 | 2.59 | 2.95 | 2.82 | 2.91 | 2.72 | 2.64 | 2.80 |
| XJ | 5.61 | 5.61 | 5.50 | 5.29 | 5.17 | 4.83 | 5.22 | 4.91 | 4.83 | 5.99 | 5.97 | 5.93 |
| Province | (b) 2020 (%) | | | | | | | | | | | |
| | Jan | Feb | Mar | April | May | Jun | Jul | Aug | Sep | Oct | Nov | Dec |
| BJ | 1.07 | 1.07 | 0.93 | 0.62 | 0.55 | 0.80 | 0.76 | 0.71 | 0.68 | 0.64 | 1.04 | 0.93 |
| TJ | 1.49 | 1.50 | 1.34 | 1.35 | 1.18 | 1.39 | 1.34 | 1.35 | 1.39 | 1.55 | 1.47 | 1.43 |
| HEB | 5.80 | 5.80 | 5.16 | 4.78 | 4.74 | 5.76 | 5.24 | 4.94 | 5.03 | 5.23 | 5.15 | 5.03 |
| SX | 5.79 | 5.79 | 5.73 | 5.50 | 5.08 | 5.60 | 6.05 | 5.32 | 5.85 | 6.15 | 5.88 | 5.27 |
| IM | 9.17 | 9.17 | 8.74 | 9.09 | 8.75 | 9.04 | 9.93 | 8.90 | 9.47 | 10.03 | 8.66 | 7.76 |
| LN | 3.12 | 3.12 | 2.78 | 2.40 | 2.63 | 2.84 | 3.26 | 2.77 | 2.51 | 2.87 | 2.97 | 2.71 |
| JL | 1.72 | 1.72 | 1.46 | 1.29 | 1.28 | 1.35 | 1.59 | 1.36 | 1.24 | 1.46 | 1.40 | 1.32 |
| HLJ | 1.88 | 1.88 | 2.01 | 1.87 | 1.73 | 1.60 | 1.77 | 1.52 | 1.47 | 1.93 | 1.72 | 1.56 |
| SH | 1.80 | 1.79 | 1.52 | 1.44 | 1.41 | 1.47 | 1.35 | 1.82 | 1.38 | 1.16 | 1.34 | 1.80 |
| JS | 7.52 | 7.52 | 8.01 | 8.54 | 8.56 | 8.90 | 8.03 | 9.58 | 8.95 | 8.34 | 7.58 | 8.42 |
| ZJ | 3.07 | 3.07 | 4.67 | 4.78 | 5.58 | 5.27 | 4.79 | 5.45 | 4.86 | 4.73 | 4.60 | 4.40 |
| AH | 4.62 | 4.62 | 5.03 | 4.57 | 4.68 | 4.82 | 4.43 | 5.17 | 4.90 | 4.92 | 4.71 | 5.07 |
| FJ | 2.27 | 2.27 | 2.64 | 2.68 | 2.75 | 3.06 | 3.90 | 3.69 | 3.54 | 3.16 | 2.83 | 2.63 |
| JX | 1.90 | 1.90 | 1.91 | 1.94 | 2.20 | 2.13 | 2.40 | 2.52 | 2.29 | 2.28 | 2.29 | 2.33 |
| SD | 9.91 | 9.91 | 10.15 | 9.50 | 8.68 | 9.65 | 9.29 | 9.25 | 10.81 | 11.14 | 9.37 | 8.64 |
| HEN | 4.99 | 4.99 | 4.49 | 4.11 | 4.77 | 5.10 | 4.91 | 4.99 | 4.68 | 4.49 | 4.48 | 4.49 |
| HUB | 2.41 | 2.40 | 1.54 | 2.27 | 2.54 | 2.34 | 1.75 | 2.50 | 2.20 | 1.89 | 2.50 | 2.93 |
| HUN | 1.69 | 1.69 | 1.24 | 1.40 | 1.59 | 1.35 | 1.65 | 1.98 | 1.40 | 1.19 | 1.87 | 2.08 |
| GD | 4.71 | 4.71 | 6.30 | 6.61 | 8.34 | 7.68 | 8.89 | 6.84 | 7.66 | 6.59 | 6.46 | 6.09 |
| GX | 2.19 | 2.19 | 2.00 | 2.25 | 2.67 | 1.73 | 1.64 | 1.80 | 1.92 | 1.73 | 2.10 | 2.10 |
| HAN | 0.41 | 0.41 | 0.43 | 0.40 | 0.50 | 0.43 | 0.47 | 0.35 | 0.46 | 0.36 | 0.25 | 0.28 |





| | | | | | | | | | | | | |
|---|---|---|---|---|---|---|---|---|---|---|---|---|
| CQ | 1.11 | 1.11 | 1.06 | 1.14 | 1.15 | 0.90 | 0.69 | 1.18 | 0.76 | 0.62 | 1.03 | 1.35 |
| SC | 1.34 | 1.34 | 0.88 | 1.27 | 1.21 | 0.89 | 0.69 | 0.68 | 0.63 | 0.65 | 0.92 | 1.42 |
| GZ | 2.20 | 2.20 | 2.72 | 3.12 | 3.01 | 1.90 | 1.30 | 2.49 | 2.04 | 1.99 | 3.28 | 3.44 |
| YN | 0.86 | 0.86 | 1.16 | 1.29 | 1.28 | 0.99 | 0.43 | 0.30 | 0.32 | 0.43 | 0.61 | 0.98 |
| Tibet | 0.01 | 0.01 | 0.01 | 0.01 | 0.01 | 0.01 | 0.01 | 0.01 | 0.01 | 0.01 | 0.01 | 0.01 |
| SAX | 4.33 | 4.33 | 4.09 | 4.12 | 3.71 | 3.37 | 3.44 | 3.20 | 3.64 | 3.69 | 3.87 | 3.95 |
| GS | 2.20 | 2.20 | 2.19 | 1.87 | 1.36 | 1.32 | 1.18 | 1.26 | 1.08 | 1.23 | 1.94 | 2.02 |
| QH | 0.36 | 0.36 | 0.40 | 0.27 | 0.13 | 0.06 | 0.06 | 0.05 | 0.08 | 0.13 | 0.21 | 0.31 |
| NX | 3.12 | 3.12 | 3.07 | 3.03 | 2.76 | 2.61 | 2.93 | 2.72 | 2.66 | 3.03 | 3.02 | 2.94 |
| XJ | 6.95 | 6.95 | 6.37 | 6.47 | 5.15 | 5.64 | 5.82 | 5.31 | 6.08 | 6.39 | 6.43 | 6.32 |

**Table 4. Provincial monthly weight factor from the industry sector in 2019 (a) and 2020 (b).**

| Province | (a) 2019 (%) | | | | | | | | | | | |
|---|---|---|---|---|---|---|---|---|---|---|---|---|
| | Jan | Feb | Mar | April | May | Jun | Jul | Aug | Sep | Oct | Nov | Dec |
| BJ | 0.12 | 0.12 | 0.13 | 0.15 | 0.19 | 0.18 | 0.17 | 0.13 | 0.09 | 0.09 | 0.14 | 0.11 |
| TJ | 0.20 | 0.20 | 0.27 | 0.34 | 0.37 | 0.30 | 0.33 | 0.25 | 0.26 | 0.28 | 0.32 | 0.25 |
| HEB | 2.80 | 2.80 | 4.05 | 4.79 | 5.38 | 5.16 | 4.93 | 4.87 | 4.57 | 3.92 | 4.30 | 3.29 |
| SX | 0.88 | 0.88 | 1.76 | 2.30 | 2.49 | 2.73 | 2.56 | 2.46 | 2.30 | 2.17 | 2.21 | 1.57 |
| IM | 0.13 | 0.13 | 0.63 | 1.47 | 1.77 | 2.05 | 2.06 | 2.14 | 2.17 | 1.90 | 0.90 | 0.27 |
| LN | 0.27 | 0.27 | 1.14 | 2.17 | 2.20 | 2.45 | 2.49 | 2.24 | 2.73 | 2.57 | 1.82 | 1.23 |
| JL | 0.06 | 0.06 | 0.44 | 0.64 | 0.93 | 1.21 | 1.04 | 1.02 | 1.14 | 1.13 | 0.40 | 0.08 |
| HLJ | 0.06 | 0.06 | 0.36 | 0.64 | 0.95 | 1.39 | 1.21 | 1.10 | 1.66 | 1.31 | 0.52 | 0.08 |
| SH | 0.23 | 0.23 | 0.19 | 0.21 | 0.19 | 0.18 | 0.17 | 0.17 | 0.15 | 0.18 | 0.17 | 0.21 |
| JS | 8.43 | 8.43 | 7.20 | 6.33 | 6.34 | 6.62 | 6.52 | 6.57 | 5.96 | 6.61 | 6.99 | 7.44 |
| ZJ | 5.84 | 5.84 | 5.89 | 5.90 | 5.70 | 5.28 | 5.20 | 5.57 | 5.57 | 6.06 | 6.13 | 6.43 |
| AH | 6.50 | 6.50 | 6.64 | 5.79 | 5.66 | 5.36 | 5.60 | 5.70 | 5.95 | 6.08 | 6.42 | 6.50 |
| FJ | 5.77 | 5.77 | 4.11 | 3.66 | 3.39 | 3.74 | 3.75 | 3.87 | 4.06 | 4.09 | 3.80 | 4.47 |
| JX | 4.49 | 4.49 | 3.56 | 3.71 | 3.82 | 3.57 | 3.83 | 4.34 | 4.41 | 4.62 | 4.37 | 4.82 |
| SD | 4.64 | 4.64 | 6.59 | 6.50 | 6.87 | 6.46 | 6.41 | 5.55 | 6.07 | 6.14 | 6.95 | 5.49 |
| HEN | 4.48 | 4.48 | 5.36 | 5.45 | 5.30 | 4.38 | 5.19 | 5.04 | 3.82 | 3.80 | 3.82 | 3.35 |
| HUB | 5.34 | 5.34 | 5.49 | 4.92 | 4.67 | 4.51 | 4.94 | 4.79 | 5.07 | 4.15 | 5.07 | 5.83 |
| HUN | 4.57 | 4.57 | 4.47 | 4.39 | 4.22 | 4.45 | 4.70 | 5.02 | 5.11 | 4.94 | 5.09 | 5.65 |
| GD | 9.33 | 9.33 | 6.99 | 6.46 | 6.16 | 6.21 | 6.42 | 6.33 | 6.73 | 7.15 | 7.74 | 8.99 |
| GX | 6.72 | 6.72 | 5.31 | 4.65 | 4.76 | 4.90 | 4.52 | 4.72 | 4.92 | 5.40 | 5.37 | 6.20 |
| HAN | 0.91 | 0.91 | 0.92 | 0.85 | 0.78 | 0.79 | 0.83 | 0.71 | 0.71 | 0.85 | 0.97 | 1.13 |
| CQ | 4.13 | 4.13 | 3.20 | 2.90 | 2.57 | 2.39 | 2.52 | 2.61 | 2.68 | 2.58 | 3.01 | 3.49 |
| SC | 8.76 | 8.76 | 6.94 | 6.40 | 5.77 | 5.56 | 5.30 | 5.30 | 5.26 | 5.47 | 5.76 | 6.57 |
| GZ | 4.73 | 4.73 | 5.04 | 4.71 | 4.41 | 4.47 | 4.46 | 4.64 | 4.37 | 4.24 | 4.90 | 5.67 |
| YN | 7.18 | 7.18 | 6.54 | 5.43 | 5.29 | 5.46 | 4.75 | 4.77 | 4.48 | 4.90 | 5.45 | 6.78 |
| Tibet | 0.31 | 0.31 | 0.32 | 0.45 | 0.56 | 0.64 | 0.68 | 0.53 | 0.57 | 0.52 | 0.35 | 0.14 |
| SAX | 2.28 | 2.28 | 2.92 | 3.35 | 3.17 | 3.13 | 2.99 | 2.97 | 2.72 | 2.76 | 2.68 | 2.37 |
| GS | 0.49 | 0.49 | 1.63 | 2.21 | 2.14 | 2.40 | 2.30 | 2.38 | 2.22 | 2.25 | 1.79 | 0.90 |





| | | | | | | | | | | | |
|---|---|---|---|---|---|---|---|---|---|---|---|
| QH | 0.12 | 0.12 | 0.30 | 0.60 | 0.70 | 0.79 | 0.82 | 0.87 | 0.79 | 0.64 | 0.45 | 0.21 |
| NX | 0.11 | 0.11 | 0.71 | 0.90 | 0.99 | 1.00 | 1.06 | 1.03 | 1.01 | 1.09 | 0.75 | 0.24 |
| XJ | 0.13 | 0.13 | 0.90 | 1.76 | 2.24 | 2.23 | 2.25 | 2.30 | 2.44 | 2.07 | 1.36 | 0.25 |

| Province | (b) 2020 (%) | | | | | | | | | | | |
|---|---|---|---|---|---|---|---|---|---|---|---|---|
| | Jan | Feb | Mar | April | May | Jun | Jul | Aug | Sep | Oct | Nov | Dec |
| BJ | 0.06 | 0.06 | 0.02 | 0.09 | 0.11 | 0.15 | 0.15 | 0.14 | 0.11 | 0.14 | 0.15 | 0.15 |
| TJ | 0.21 | 0.21 | 0.09 | 0.23 | 0.24 | 0.25 | 0.26 | 0.19 | 0.22 | 0.24 | 0.29 | 0.22 |
| HEB | 2.92 | 2.92 | 4.09 | 4.75 | 5.13 | 5.40 | 5.42 | 5.45 | 5.44 | 5.32 | 5.06 | 4.19 |
| SX | 0.63 | 0.63 | 2.11 | 2.35 | 2.42 | 2.69 | 2.60 | 2.46 | 2.80 | 2.49 | 2.16 | 1.38 |
| IM | 0.17 | 0.17 | 0.45 | 1.20 | 1.79 | 2.06 | 2.19 | 2.23 | 2.22 | 1.92 | 1.03 | 0.29 |
| LN | 0.51 | 0.51 | 1.25 | 2.36 | 2.34 | 2.67 | 2.71 | 2.64 | 2.62 | 2.69 | 2.30 | 1.53 |
| JL | 0.07 | 0.07 | 0.30 | 0.67 | 0.94 | 1.21 | 1.22 | 1.26 | 1.05 | 1.20 | 0.54 | 0.15 |
| HLJ | 0.03 | 0.03 | 0.06 | 0.57 | 0.99 | 1.40 | 1.49 | 1.49 | 1.28 | 1.35 | 0.51 | 0.09 |
| SH | 0.16 | 0.16 | 0.08 | 0.15 | 0.17 | 0.15 | 0.13 | 0.19 | 0.16 | 0.18 | 0.17 | 0.22 |
| JS | 6.42 | 6.42 | 6.46 | 6.33 | 6.26 | 5.64 | 5.56 | 6.30 | 6.35 | 6.58 | 6.93 | 8.05 |
| ZJ | 5.44 | 5.44 | 5.64 | 5.88 | 5.48 | 4.93 | 5.27 | 5.84 | 5.12 | 5.74 | 6.09 | 6.11 |
| AH | 6.90 | 6.90 | 7.00 | 6.09 | 5.89 | 5.09 | 4.75 | 5.60 | 6.05 | 5.96 | 6.21 | 7.04 |
| FJ | 5.68 | 5.68 | 4.69 | 3.55 | 3.47 | 3.83 | 4.08 | 4.07 | 3.96 | 4.02 | 3.87 | 4.71 |
| JX | 5.33 | 5.33 | 3.98 | 3.87 | 3.89 | 3.51 | 3.74 | 4.26 | 3.78 | 4.18 | 4.37 | 4.68 |
| SD | 4.98 | 4.98 | 6.87 | 7.05 | 6.88 | 6.91 | 6.93 | 6.16 | 7.04 | 7.13 | 7.01 | 4.71 |
| HEN | 4.55 | 4.55 | 6.53 | 5.92 | 5.49 | 4.85 | 5.52 | 4.99 | 5.00 | 4.24 | 4.19 | 3.42 |
| HUB | 3.76 | 3.76 | 1.43 | 3.72 | 4.25 | 4.01 | 3.97 | 4.77 | 4.63 | 4.35 | 4.87 | 5.75 |
| HUN | 5.73 | 5.73 | 5.09 | 4.51 | 4.32 | 4.25 | 4.36 | 4.48 | 4.24 | 4.34 | 4.79 | 5.64 |
| GD | 9.66 | 9.66 | 7.07 | 6.53 | 6.26 | 6.14 | 6.95 | 6.49 | 6.60 | 6.92 | 7.76 | 9.35 |
| GX | 7.05 | 7.05 | 5.65 | 4.76 | 4.79 | 4.72 | 4.86 | 4.59 | 4.64 | 4.87 | 5.16 | 6.25 |
| HAN | 0.85 | 0.85 | 0.65 | 0.75 | 0.78 | 0.77 | 0.85 | 0.70 | 0.71 | 0.61 | 0.83 | 1.03 |
| CQ | 3.65 | 3.65 | 3.05 | 2.74 | 2.78 | 2.61 | 2.25 | 2.61 | 2.61 | 2.39 | 2.78 | 3.15 |
| SC | 9.13 | 9.13 | 7.76 | 6.39 | 5.93 | 6.01 | 5.43 | 4.96 | 5.16 | 5.23 | 6.02 | 6.88 |
| GZ | 5.58 | 5.58 | 5.75 | 5.03 | 4.32 | 4.42 | 4.38 | 4.57 | 3.67 | 3.83 | 4.50 | 4.81 |
| YN | 7.15 | 7.15 | 6.89 | 5.55 | 5.28 | 5.58 | 5.09 | 4.75 | 4.55 | 4.64 | 5.25 | 6.50 |
| Tibet | 0.11 | 0.11 | 0.26 | 0.42 | 0.48 | 0.71 | 0.57 | 0.60 | 0.56 | 0.46 | 0.32 | 0.30 |
| SAX | 2.38 | 2.38 | 3.43 | 3.13 | 3.00 | 2.99 | 2.89 | 2.92 | 3.25 | 2.90 | 2.82 | 1.95 |
| GS | 0.58 | 0.58 | 1.73 | 2.40 | 2.29 | 2.50 | 2.29 | 2.25 | 2.25 | 2.10 | 1.71 | 0.85 |
| QH | 0.07 | 0.07 | 0.26 | 0.49 | 0.56 | 0.72 | 0.66 | 0.71 | 0.67 | 0.64 | 0.43 | 0.19 |
| NX | 0.10 | 0.10 | 0.63 | 0.90 | 1.00 | 1.05 | 1.09 | 1.01 | 1.05 | 1.02 | 0.71 | 0.21 |
| XJ | 0.14 | 0.14 | 0.72 | 1.62 | 2.49 | 2.79 | 2.32 | 1.32 | 2.20 | 2.31 | 1.17 | 0.20 |

**Table 5. Provincial weight factor from the industry sector. WF represents weight factor.**

| Province | WF (%) | Province | WF (%) | Province | WF (%) |
|---|---|---|---|---|---|
| BJ | 2.47 | ZJ | 6.60 | HAN | 0.55 |
| TJ | 1.29 | AH | 3.50 | CQ | 1.80 |
| HEB | 6.59 | FJ | 2.68 | SC | 4.73 |
| SX | 2.81 | JX | 2.31 | GZ | 2.06 |



| | | | | | |
|---|---|---|---|---|---|
| IM | 2.29 | SD | 9.16 | YN | 2.92 |
| LN | 3.43 | GZ | 6.24 | Tibet | 0.22 |
| JL | 1.82 | HUB | 3.32 | SAX | 2.66 |
| HLJ | 2.06 | HUN | 3.36 | GS | 1.36 |
| SH | 1.69 | GD | 9.11 | QH | 0.47 |
| JS | 7.65 | GX | 2.53 | NX | 0.62 |
| XJ | 1.71 | | | | |

**Table 6. The duration of holidays in 2019 and 2020.**

| | 2019 | 2020 |
|---|---|---|
| New Year | 1 (1st Jan) | 1 (1st Jan) |
| Spring Festival | 7 (4th-10th Feb) | 10 (24th Jan-2nd Feb) |
| Qingming Festival | 3 (5th-7th Apr) | 3 (4th-6th Apr) |
| Labor Day | 4 (1st-4th May) | 5 (1st-5th May) |
| Duanwu Festival | 3 (6th-8th Jun) | 3 (25th-27th Jun) |
| Mid-autumn Festival | 3 (13th-15th Sep) | |
| National Day | 7 (1st-7th Oct) | |
| Mid-autumn Festival+National Day | | 8 (1st-8th Oct) |

**Table 7. The maximum and minimum average daily CO$_2$ emissions during a monthly period.** Max-value and Min-value (unit in thousand tons of CO$_2$) represent the maximum and minimum average daily CO$_2$ emissions from January to December, respectively. Max-month and Min-month represent the month of Max-value and Min-value, respectively.

| Province | 2019 | | | | 2020 | | | |
|---|---|---|---|---|---|---|---|---|
| | Max-value | Max-month | Min-value | Min-month | Max-value | Max-month | Min-value | Min-month |
| BJ | 218.76 | Jan | 148.59 | Apr | 231.14 | Dec | 134.11 | Apr |
| TJ | 298.01 | Dec | 208.35 | Oct | 292.10 | Dec | 171.46 | Feb |
| HEB | 1504.43 | Jun | 1038.11 | Feb | 1563.70 | Jun | 816.01 | Feb |
| SX | 1179.77 | July | 795.24 | Feb | 1172.00 | July | 628.09 | Feb |
| IM | 1546.09 | Aug | 1021.28 | Feb | 1628.95 | Aug | 918.07 | Feb |
| LN | 796.29 | July | 442.85 | Feb | 833.84 | July | 371.05 | Feb |
| JL | 358.82 | July | 219.25 | Feb | 398.24 | July | 190.19 | Feb |
| HLJ | 442.15 | Jun | 267.98 | Feb | 460.07 | July | 206.40 | Feb |
| SH | 327.82 | Jan | 200.96 | May | 361.51 | Dec | 194.01 | Oct |
| JS | 2248.65 | Dec | 1895.21 | Feb | 2570.50 | Dec | 1214.57 | Feb |
| ZJ | 1644.01 | Dec | 1119.25 | Feb | 1675.47 | Aug | 712.81 | Feb |
| AH | 1590.19 | Dec | 1198.58 | Feb | 1797.30 | Dec | 918.42 | Feb |
| FJ | 998.65 | Dec | 775.03 | May | 1105.20 | Aug | 606.53 | Feb |
| JX | 944.01 | Dec | 654.71 | Feb | 1025.60 | Dec | 545.07 | Feb |
| SD | 2379.96 | Nov | 1766.90 | Feb | 2477.22 | Sep | 1370.22 | Feb |
| HEN | 1562.23 | July | 1062.25 | Dec | 1493.97 | Aug | 836.75 | Feb |





| | | | | | | | | |
|---|---|---|---|---|---|---|---|---|
| HUB | 1220.24 | Dec | 817.10 | Dec | 1282.58 | Dec | 374.27 | Mar |
| HUN | 1050.76 | Dec | 669.25 | Feb | 1132.73 | Dec | 561.96 | Feb |
| GD | 2157.74 | Dec | 1512.92 | Feb | 2392.54 | Dec | 1165.01 | Feb |
| GX | 1081.54 | Dec | 762.69 | July | 1192.86 | Dec | 683.77 | Feb |
| HAN | 195.21 | Dec | 130.85 | Feb | 189.20 | Dec | 98.96 | Feb |
| CQ | 639.08 | Dec | 432.79 | Jun | 661.45 | Dec | 357.05 | Feb |
| SC | 1097.89 | Dec | 818.46 | July | 1215.52 | Dec | 758.39 | Feb |
| GZ | 1158.96 | Dec | 707.93 | Feb | 1216.74 | Dec | 586.51 | Feb |
| YN | 1001.56 | Dec | 653.67 | Sep | 1050.60 | Dec | 566.12 | Feb |
| Tibet | 85.26 | July | 22.95 | Dec | 95.88 | Jun | 10.27 | Feb |
| SAX | 932.69 | Dec | 672.54 | Feb | 978.21 | Nov | 596.12 | Feb |
| GS | 491.43 | Nov | 317.21 | Feb | 534.28 | Apr | 262.71 | Feb |
| QH | 121.99 | Aug | 68.17 | Feb | 109.87 | Jun | 44.20 | Feb |
| NX | 529.20 | July | 351.58 | Feb | 536.81 | Aug | 312.74 | Feb |
| XJ | 995.69 | July | 705.71 | Feb | 1102.21 | Jun | 696.57 | Feb |

688

**Table 8. The maximum and minimum daily CO2 emissions and the date of maximum and minimum daily CO2 emissions for the years 2019 and 2020.**

| Province | 2019 | | | | 2020 | | | |
|---|---|---|---|---|---|---|---|---|
| | max | max-date | min | min-date | max | max-date | min | min-date |
| BJ | 238.47 | 2nd Jan | 136.50 | 1st May | 259.44 | 25th Nov | 108.19 | 4th Apr |
| TJ | 311.88 | 23rd Dec | 169.37 | 5th Feb | 322.38 | 25th Nov | 157.80 | 9th Feb |
| HEB | 1644.33 | 26th Jul | 833.74 | 5th Feb | 1833.70 | 25th Nov | 750.25 | 9th Feb |
| SX | 1321.05 | 26th Jul | 624.80 | 5th Feb | 1390.64 | 25th Nov | 579.48 | 13rd Feb |
| IM | 1729.28 | 26th Jul | 794.23 | 5th Feb | 1760.15 | 30th Jul | 846.90 | 13rd Feb |
| LN | 887.56 | 26th Jul | 359.50 | 5th Feb | 947.89 | 25th Nov | 340.47 | 9th Feb |
| JL | 399.22 | 26th Jul | 178.72 | 5th Feb | 430.42 | 30th Jul | 174.42 | 9th Feb |
| HLJ | 481.52 | 9th Sep | 217.43 | 5th Feb | 513.57 | 22nd Jan | 189.16 | 9th Feb |
| SH | 362.96 | 2nd Jan | 177.40 | 1st May | 370.87 | 30th Dec | 133.79 | 2nd Jan |
| JS | 2462.56 | 29th Nov | 1494.95 | 5th Feb | 2638.44 | 30th Dec | 1120.04 | 9th Feb |
| ZJ | 1769.28 | 29th Nov | 895.79 | 5th Feb | 1906.72 | 25th Nov | 654.05 | 9th Feb |
| AH | 1739.33 | 29th Nov | 937.18 | 5th Feb | 1865.33 | 25th Nov | 847.29 | 13rd Feb |
| FJ | 1064.38 | 9th Sep | 619.51 | 5th Feb | 1189.15 | 17th Aug | 559.58 | 13rd Feb |
| JX | 1022.51 | 29th Nov | 514.41 | 5th Feb | 1138.33 | 25th Nov | 502.87 | 13rd Feb |
| SD | 2746.37 | 29th Nov | 1406.35 | 5th Feb | 2910.49 | 25th Nov | 1262.75 | 9th Feb |
| HEN | 1742.59 | 26th Jul | 901.54 | 1st Oct | 1604.27 | 17th Aug | 770.12 | 9th Feb |
| HUB | 1279.00 | 23rd Dec | 669.19 | 5th Feb | 1316.63 | 30th Dec | 321.87 | 1st Mar |
| HUN | 1124.05 | 29th Nov | 532.01 | 5th Feb | 1164.44 | 25th Nov | 510.31 | 2nd Oct\ |
| GD | 2302.20 | 29th Nov | 1212.04 | 5th Feb | 2545.89 | 25th Nov | 1071.58 | 9th Feb |
| GX | 1216.50 | 29th Nov | 624.59 | 5th Feb | 1240.07 | 25th Nov | 576.60 | 2nd Oct |
| HAN | 204.60 | 23rd Dec | 103.33 | 5th Feb | 194.20 | 30th Dec | 86.71 | 2nd Oct |
| CQ | 670.97 | 2nd Jan | 369.71 | 1st Oct | 678.98 | 30th Dec | 277.90 | 2nd Oct |



| | | | | | | | |
|---|---|---|---|---|---|---|---|
| SC | 1213.74 | 2nd Jan | 695.89 | 1st Oct | 1274.44 | 9th Oct | 571.12 | 2nd Oct |
| GZ | 1215.89 | 23rd Dec | 553.49 | 5th Feb | 1316.95 | 25th Nov | 501.27 | 2nd Oct |
| YN | 1049.60 | 23rd Dec | 549.25 | 5th Feb | 1078.43 | 30th Dec | 470.26 | 2nd Oct |
| Tibet | 95.43 | 26th Jul | 21.42 | 1st Dec | 101.86 | 15th Jun | 9.23 | 9th Feb |
| SAX | 1043.79 | 29th Nov | 530.14 | 5th Feb | 1160.70 | 25th Nov | 549.98 | 13rd Feb |
| GS | 569.14 | 29th Nov | 250.69 | 5th Feb | 630.91 | 25th Nov | 242.39 | 13rd Feb |
| QH | 133.49 | 9th Sep | 55.00 | 5th Feb | 125.07 | 22nd Oct | 40.45 | 9th Feb |
| NX | 594.27 | 26th Jul | 272.39 | 5th Feb | 629.72 | 25th Nov | 288.48 | 13rd Feb |
| XJ | 1143.20 | 29th Nov | 549.59 | 5th Feb | 1296.39 | 25th Nov | 642.57 | 13rd Feb |

692

**Table 9. Reduced emissions over the weekend and sectoral contributions to reduced emissions over the weekend.**

696

| Province | Emissions (thousand tons $CO_2$) | | | Contribution rate (%) | | |
|---|---|---|---|---|---|---|
| | Power | Industry | Transport | Power | Industry | Transport |
| HLJ | 5.38 | 1.85 | 6.09 | 40.39 | 13.92 | 45.69 |
| XJ | 19.57 | 4.57 | 5.06 | 67.02 | 15.66 | 17.33 |
| SX | 18.66 | 5.09 | 8.32 | 58.2 | 15.87 | 25.93 |
| NX | 9.06 | 1.88 | 1.85 | 70.87 | 14.69 | 14.44 |
| Tibet | 0.02 | 1.39 | 0.66 | 1.05 | 67.17 | 31.78 |
| SD | 30.81 | 14.8 | 27.14 | 42.35 | 20.35 | 37.3 |
| HEN | 15.67 | 13.07 | 18.49 | 33.19 | 27.67 | 39.14 |
| JS | 26.15 | 16.91 | 22.65 | 39.8 | 25.73 | 34.48 |
| AH | 15.03 | 14.72 | 10.38 | 37.44 | 36.68 | 25.87 |
| HUB | 7.37 | 11.13 | 9.85 | 25.99 | 39.26 | 34.74 |
| ZJ | 12.34 | 14 | 19.55 | 26.9 | 30.51 | 42.6 |
| JX | 6.74 | 10.95 | 6.85 | 27.46 | 44.6 | 27.93 |
| HUN | 4.99 | 12.39 | 9.96 | 18.26 | 45.31 | 36.43 |
| YN | 1.92 | 14.46 | 8.64 | 7.68 | 57.79 | 34.53 |
| GZ | 6.01 | 11.46 | 6.11 | 25.48 | 48.61 | 25.91 |
| FJ | 9.18 | 11.7 | 7.94 | 31.86 | 40.59 | 27.55 |
| GX | 5.45 | 14.25 | 7.5 | 20.02 | 52.39 | 27.59 |
| GD | 19.94 | 20.52 | 26.99 | 29.57 | 30.42 | 40.01 |
| HAN | 1.35 | 2.33 | 1.62 | 25.54 | 43.92 | 30.54 |
| JL | 4.44 | 1.67 | 5.38 | 38.63 | 14.51 | 46.86 |
| LN | 9.02 | 5.28 | 10.15 | 36.88 | 21.59 | 41.52 |
| TJ | 4.27 | 0.74 | 3.81 | 48.4 | 8.42 | 43.18 |
| QH | 0.6 | 1.39 | 1.4 | 17.77 | 40.95 | 41.27 |
| GS | 4.5 | 4.43 | 4.02 | 34.75 | 34.22 | 31.03 |
| SAX | 10.95 | 6.95 | 7.87 | 42.51 | 26.97 | 30.53 |
| IM | 28.1 | 3.67 | 6.78 | 72.89 | 9.52 | 17.59 |
| CQ | 2.8 | 7.46 | 5.34 | 17.94 | 47.8 | 34.26 |





| | | | | | | |
|---|---|---|---|---|---|---|
| **HEB** | 16.95 | 11.06 | 19.51 | 35.67 | 23.28 | 41.05 |
| **SH** | 4.73 | 0.46 | 5.02 | 46.36 | 4.52 | 49.12 |
| **BJ** | 2.58 | 0.36 | 7.32 | 25.14 | 3.47 | 71.39 |
| **SC** | 2.95 | 17.42 | 14 | 8.58 | 50.68 | 40.74 |