# Peer review of "Daily CO2 emission for China's provinces in 2019 and 2020"

_Earth System Science Data, 2021_

## Author Comment (AC1)

We sincerely thank you for the valuable feedback that we have used to improve the quality of our manuscript. According to the comments from you and another reviewer, we re-estimated daily provincial CO2 emissions use more proxies: 1) For power sector, we added yearly provincial thermal power generation data to existing monthly provincial thermal power generation; 2) for industry sector, we re-collected monthly provincial productions of 35 industrial products to replace the previous monthly cement productions (Table 1); 3) for ground transport sector, except for existing yearly vehicles ownership, we added 3 monthly proxies, one seasonal proxy, and additional 7 yearly proxies. In addition, we added the comparison with the other two additional dependent China provincial carbon emissions datasets except for existing CEADs. Moreover, we re-estimate our uncertainties. I think these improvements could address your concern about using only cement production to estimate emissions from industry sector. Below you will find out point-by-point responses to the reviewers' comments/questions.

*Q1: Line 49: Need to expand and provide citations for EDGAR, CDIAC and CEADs here in the introduction.*

**Response:** Thanks for your comments. The reference of EDGAR (Crippa et al., 2019), CDIAC (Boden et al., 2016), and CEADs (Shan et al., 2018) are as follows:

Crippa, M., Oreggioni, G., Guizzardi, D., Muntean, M., Schaaf, E., Lo Vullo, E., Solazzo, E., Monforti-Ferrario, F., Olivier, J.G.J., Vignati, E., Fossil CO2 and GHG emissions of all world countries - 2019 Report, EUR 29849 EN, Publications Office of the European Union, Luxembourg, 2019, ISBN 978-92-76-11100-9, doi:10.2760/687800, JRC117610.
Boden, T.A., G. Marland, and R.J. Andres. 2016. Global, Regional, and National Fossil-Fuel CO2 Emissions. Carbon Dioxide Information Analysis Center, Oak Ridge National Laboratory, U.S. Department of Energy, Oak Ridge, Tenn., U.S.A. doi 10.3334/CDIAC/00001_V2016
Shan Y,Guan D,Zheng H,et al. China CO2 emission accounts 1997–2015[J]. Scientific Data, 2018, 5:170201.

*Q2: What is China's mitigation target, by what date and according to what baseline year?*

**Response:** In 2015 China committed to improving the carbon intensity by 40% - 45% relative to 2005 levels by 2020, and the peak of total emission by 2030 (Goverment, 2015). In 2020, China's government announced the ahead achievement of the carbon intensity target by decreasing 48.1% in 2019 relative to 2005 levels, and the share of renewables achieved 15.3% in the energy mix (Office, 2020). In September 2020, Chinese President Xi Jinping announced China's new climate target to peak its $CO_2$ emissions before 2030 and to achieve carbon neutrality before 2060 (Jinping, 2020).

Goverment, T.C. (2015) Enhanced Actions on Climate Change: China's Intended Nationally Determined Contributions.
Office, C.s.S.C.I. (2020) Energy in China's New Era.

Jinping, X. (2020) Statement by H.E. Xi Jinping President of the People's Republic of China At the General Debate of the 75th Session of The United Nations General Assembly.

*Q3: Line 76: Do the authors mean "by acknowledging the uncertainties more than just the total emissions"?*

**Response:** Thanks for your comments. We replaced the "inventory" in the last manuscript with total emissions.

*Q4: Line 152: by "totally" do the authors mean "combined"? If so changing to combined would be clearer.*

**Response:** Thanks for your comments. We replaced the "totally" in the last manuscript with "combined".

*Q5: Line 170: remove "were"*

**Response:** Thanks for your comments. But we re-write this sector, the "were" has been removed. The re-write sector is as follows:

For the industry sector, we collected monthly provincial production of 35 industrial products related to emissions sectors to construct the provincial weight factor of industrial $CO_2$ emissions. By reviewing the subsectors of the industrial sector at the national level of China as presented in the EDGAR2019 and CEADs dataset, we found that the emission in manufacturing sector accounts for almost 90% of the manufacturing industries and construction sector (Table 3). We have also had the industrial sector in China subdivided into 10 subsectors in accordance with the EDGAR classification standard and also collected the provincial monthly data of 35 types of products relevant to emission (Table. 3). Only 9 industrial subsectors are covered, considering that the 35 types of products are irrelevant to the subsector of wood and wood products. According to the results of EDGAR (2019), the emission of the subsector of wood and wood products only accounts for 0.11% of the total emission of the handicraft industry. Therefore, it is believed that the 35 types of industrial products cover nearly 90% of the industrial emission nevertheless, and can reflect the overall situation of industrial emission. The provincial daily emissions of the industrial sector in China were estimated on the basis of those data and the following equation:

$$E_{pI} = \sum_{m} (E_{NI} \times R_m \times (\sum_{n} \frac{p_{p,n}}{p_{t,n}}) * 1/n)$$

where $E_{pI}$ is the provincial emission of industrial sector; $E_{NI}$ is the national emission of industrial sector; m is the subsector of the industrial sector, where m=9; $R_m$ corresponds to the contribution rate of the subsector of the industrial sector to the total emission of handicraft industry; n is the number of the relevant product in the subsector; $P_{p,n}$ represents the provincial production of the product; and $P_{t,n}$ represents the total production of the product. It is reckoned that the estimation made in this method can cover nearly 90% of the nationwide emission.

*Q6: Line 217: Should refer to Figure 1 regarding the "valley shape" which needs a few months to illustrate the shape. Table 1 lists the minimum and maximum only.*

**Response:** Thanks for your comments. For more clear representation, we used Table 1 lists the average monthly emissions at the provincial level.

*Q7: For most readers who will have a limited knowledge of Chinese provinces, the lists of province abbreviations is tedious to read and not useful, even with the Table 1 legend. I acknowledge that perhaps this comment is unwarranted and just my own bias, since I would not feel the same if it were my own country's sub-national regions or those of the US, which are more commonly known. My suggestion is to just state the number of provinces in the text, rather than provide long lists, since the tables and figures give that detail anyway. A compromise to improve readability may be to just state the number and list them in parentheses. I will leave this to the authors and editor to decide on what is best.*

**Response:** Thanks for your comments. We added the number column in Table 1, and in provincial figures, we used numbers instead of letters in figure titles (like new Figure 1). The provinces in all provincial figures use the same numerical order. Therefore, we could use numbers to represent the long name of provinces.

*Q8: It should be stated that "1 ton = 1000 kg". In some usage, like in the US, "ton" is 2000 pounds, while "tonne" or "metric ton" is 1000 kg.*

**Response:** Thanks for your comments. We replaced tonne with "metric tons", please see Figure 1.

*Q9: Line 386: "which fell during the COVID-19 pandemic" should be replaced with " which fell during a period of national restrictions due to the COVID-19 pandemic", since the pandemic is still going on.*

**Response:** Thanks for your comments. We replaced "which fell during the COVID-19 pandemic" with "which fell during a period of national restrictions due to the COVID-19 pandemic".

*Q10: Figure 1-6. Clearer and larger x-axis labels would be an improvement.*

**Response:** Thanks for your comments. We have improved larger x-axis like new Figure 1.

*Q11: Figures 1-11. Listing the provincial abbreviations in the figure captions is not helpful and they should be removed.*

**Response:** Thanks for your comments. We deleted all provincial abbreviations in the figure captions.

*Q12: Figure 5 says southern China, but it is actually all 31 provinces.*

**Response:** Thanks for your comments. We deleted "southern" in the Figure 5 captions

*Q13: Table 3 and 4. It might be helpful to note that the sum of each column or month is 100%.*

**Response:** Thanks for your comments. We added 100% under the month title in each column.

[Figure]

**Figure 1.** Total daily CO$_2$ emissions from the Chinese mainland's 31 provinces from 2019 to 2020. Total emissions cover 3 sectors: electricity (green), industry (orange) and ground transport (blue).

---

## Author Comment (AC2)

Thank you very much for your insightful comments, which are helpful to improve the quality of this research data. Based on your comments, the provincial distribution method in industrial sector and that in ground transport sector have been improved; an additional comparison between this dataset and another two databases (MEIC and PKU) at the provincial level of China has been added, and uncertainties of provincial daily emission in industrial sector and ground transport sector have been re-estimated. The revised parts that correspond to the comments are listed as follows:

Q1. The authors state in their introduction that "(current) estimates of China's CO2 emissions carry significant uncertainties" and "having timely and accurate CO2 emission estimates ... is therefore a fundamental prerequisite to designing evidence-based policies for reducing China's CO2 emissions." But the paper does not demonstrate that the new estimates are of any lower uncertainty than those already found. Indeed, the methododology used introduces very significant uncertainties with numerous assumptions and proxies. Can the authors argue that this new dataset has lower uncertainty than previous estimates at the provincial level?

**Response:**

This study addresses to the timelyness issue of current inventories of CO2 emissions. In order to improve this issue, we had to use numerous assumptions and proxies as your comments pointed out. These assumptions and proxies lead to higher uncertainty than other previous inventories.

We compared our results with the other three datasets of china's provincial CO2 emissions, such as CEADs, MEIC, and PKU. Referring to CEADs (Shan et al., 2018), the uncertainties ranges in national CO2 emissions during the period of 2010 and 2015 were between -15% and 25% with a confidence interval of 97.5%. In regard to MEIC (Zheng et al, 2018; Liu 2015), the uncertainties of national carbon emissions from power sector ranged from -15% to 16%. For PKU, the uncertainties reached 63.2% based the global grided CO2 emission map (Wang et al., 2013).

In this study, uncertainties were from three parties, provincial proxies, national daily emissions, and our assumptions. Given the uncertainties from provincial proxies, with a confidence interval of 97.5%, our uncertainties are as follows: 1) for power sector, Two provinces (2/31), BJ and JS have the lowest uncertainties at  $\pm 23\%$  and  $\pm 26\%$ , respectively; the uncertainties in 14 provinces (14/31) were between 30% and 100%, and the uncertainties in the other eight provinces (8/31) were more than 100%; 2) for industry sector, the province with the lowest uncertainties is HEN ( $\pm 16\%$ ), 16 provinces (16/31) had the uncertainties between 20% and 50%, 9 provinces (9/31) had the uncertainties between 50% and 100% and the other 4 provinces (4/31) with the largest uncertainties beyond 100% are IM, BJ, Tibet, and SH. 3) for ground transport sector, the uncertainties in more than 50% of provinces (16/31) were less than 10%, the uncertainties in 6 provinces were between 10% and 20%, and the uncertainties in the other 9 (9/31) provinces ranged between 20% and 50%.

As your mentioned in the following comments, our provincial daily variation in CO2

emissions has a strong dependence on national daily variation. Deng et al., (2021) discussed the uncertainty of power emissions is within  $(\pm 14\%)$  and the uncertainty of China's industrial sector is 20% with a confidence interval of 68%.

In this study, we assumed that no temporal variation in emission factors, which might introduce an uncertainty of 2% (Deng et al., 2021). In addition, the emissions from the ground transportation sector are estimated by assuming that the relative magnitude in car counts (and thus emissions) follow a similar relationship with TomTom congestion index in Paris. (Gensheimer et al., 2021) pointed out the uncertainty could be more than 60% using the Paris relationship between traffic congestion and traffic flow to estimate other regional traffic conditions.

Fig. 1, 2, and 3 (at the end of this response letter) show the uncertainty range of daily provincial emissions from power, industry, and ground transport, respectively. Table 1 lists uncertainties of provincial average daily carbon emissions from 2019 to 2020. Table 2 provides the basic information of the other three datasets (CEADs, MEIC, and PKU).

Q2. On line 148, the authors write "The provincial dataset constructed in this study includes daily CO2 emission data from the three main polluting sectors (power, industry and ground transport) in China, which together account for more than 90% of the total emissions." However, the data used to estimate these are for the power sector, cement production, and road transport, and together these accounted for only 55% of China's CO2 emissions in 2018 according to EDGAR v6.0. The way this sentence is worded gives the strong impression that the data used cover the majority of the total and implies robustness from this. However, it is not 90% but 55%. Please reword to reflect the true situation.

**Response:**

In the last version of the manuscript, only monthly provincial cement production was used as proxies to estimate total emissions from industry sector, which do not cover nearly 90% of China's industrial emissions well. In the new method, we collected monthly provincial production data of 35 industrial products to estimate emissions from industry sector. By comparing the subsectors of industrial sector at the national level of China as presented in the EDGAR2019 and CEADs dataset (Table 3 and Table 4), we found that these 35 products could cover 9 subsectors of manufacturing industries, which accounted for nearly 90% of the manufacturing industrial emissions (Table. 4).

We added Table 4 to show these 35 products and the 9 sub-sectors they belonged to.

Q3. Looking at the method used for road transport emissions, the national-level estimates for China in CarbonMonitor (the starting point of this study) are based on traffic congestion data tuned to traffic flows in one city: Paris. This method has been shown to have high and variable uncertainty when applied to other cities and countries (Gensheimer et al., 2021, who provide strong warnings that such

methods should be used with caution). Further, the correlation used is with traffic flows (vehicles per hour), which in turn is assumed to correlate well with total transport emissions.

**Response:**

Thank you very much for providing the study from (Gensheimer et al., 2021) for our reference. As the study discussed, it is hard to develop a generalizable relationship between the mobility data and traffic flow over all the study regions as we assumed. Using this assumption could introduce more than 60% of uncertainties to the estimation of CO2 emissions than using governmental data.

We did not collect high temporal resolution of China's governmental traffic data. We have tried to build a new relationship between traffic flow and traffic congestion from Hong Kong's data to improve the 'Paris relationship'. However, Hong Kong's traffic flow is base on a street near the airport, which not be used to establish the relationship between traffic flow and traffic congestion well. Therefore, due to governmental data limitations, we do not have a better way of assuming China's provinces have a similar relationship between traffic flow and traffic congestion with Paris.

Q4.Starting with an uncertain estimate of daily transport emissions in China, the method in this paper then further disaggregates that by using vehicle ownership data by province (annual, for the year 2018), on the assumption that the distance travelled each day in one province is very likely to be close to the share of that province's ownership of vehicles in China's total ownership. I find no discussion of how the age of the vehicle fleet might vary by province, how the share of diesel, petrol, electric might vary by province, and how driving behaviour might vary by province (for example according to wealth or urban density), to take just three examples. Nor do I find any discussion of how much passenger transport makes up of total ground transport in each province, when we know that goods transport is a large source of emissions. This method leaves me wondering whether the resulting estimates are of any use at all, but the paper takes it as given that they are. Since the details are not provided, I have looked at the data files and compared with NBS data and find that the authors have used "Possession of Civil Vehicles" (the English title), which appears to include all non-military vehicles. Please add sufficient details so that others might better understand the methods used.

**Response:**

Thanks for your comments, using vehicle ownership as provincial proxies of carbon emissions from ground transport does indeed need more discussions. In order to better get the features of emissions from ground transport at the provincial level, we expanded our proxies. We collected the provincial monthly productions of gasoline, diesel, and automobiles, the provincial monthly GDP data, as well as 8 indicators of provincial annual data relevant to ground transportation emission (Table 5). The provincial volume of passenger traffic and goods traffic has been also included in these 8 indicators. Vehicle ownership (Possession of Civil Vehicles) is also involved in this data, including the ownership of civil passenger vehicles, that of civil trucks, and that of other civil vehicles. In detail, the civil passenger vehicles include large, medium, small and mini passenger vehicles, while the civil trucks include heavy, medium, light, and mini trucks.

O5. Looking at the power sector, CarbonMonitor made use of daily power generation data to estimate daily CO2 emissions at the national level. However, daily data were only available for six power companies so had to combine with monthly national thermal generation data. Further, that method used a constant emission factor across all thermal generation in China, despite significant swings in total power generation caused by the pandemic early in 2020. While 'normal' changes in total production might not have much effect on the emission factor of total thermal generation over the country from month to month (and this is already debateable), large drops lead to the very clear possibility that high-cost production drops out first, potentially meaning inefficient power stations with higher emission factors. Whether this is the case or not in China has not been discussed. In this article, only monthly data were available at the provincial level, so the national CO2 emissions from power generation on a particular day are scaled based on the province's share of thermal generation in that month. Again, this leaves me wondering how accurate the resulting daily estimates will be, given that the entire day-to-day variation comes directly from the national estimates. I find no discussion of these issues in the article.

**Response:**

Thanks for your comments, we have to respond to this question from three aspects. First, in this study, we assumed that no discrepancies in Emission factors at the provincial level, which induced uncertainty of (-15.8%, 23.7%) (Shan et al., 2018). Second, referring to the effect of high-cost production drops on emission factor, emission factor might be increasing owing to the inefficient power stations under a lockdown of COVID-19 (Jan to May 2020), however, after May 2020, power generation resumed because COVID-19 was effectively controlled. Third, we used the following method to simply attribute the reason for the daily variation. we counted the number of holidays and Mondays among the 50 days with the highest national day-to-day variation in emissions of different sectors in 2019 and 2020 (Table 6). As the daily variation was calculated as per the equation "(Eday+1-Eday) / Eday", Monday was counted to facilitate reflecting the emission variations on Monday and Sunday. Our results showed that more than half of the Top 50 daily variation was caused by the effect of holidays or Mondays in 2019 (Table 4). Especially in transportation sector, above 88% of the Top daily variation in emission was caused by the effects of holidays and Mondays. In 2020, holidays and Mondays account for 44%, 40%, and 64% of the Top 50 daily change for power, industry, and transportation sectors, respectively (Table 4). Since there are no specific holidays for some provinces, the daily variation in emission caused by the effects of holidays and weekends was supposed to be consistent in all provinces of China. Thereby, the national daily variation can partially reflect the daily variation at the provincial level. It should be noted that our method is indeed limited in reflecting provincial daily changes, especially in power and industry sectors.

Q6. For the industry sector, CarbonMonitor used monthly data and then disaggregated these to daily data using the power sector emissions estimates. As that article puts it "assumes a linear relationship between daily electricity generation for industry and daily industry production data to compute daily industry production." Already a very strong assumption, with no discussion of how reasonable it might be. Here these very approximate national daily estimates are then disaggregated to provincial level using only monthly cement production data. Again, it's very difficult to imagine that the resulting estimates of daily industry emissions at provincial level are of high quality.

**Response:** Thanks for your comments. In the new method, we used monthly production of 35 industrial products to instant cement production in order to better estimate industrial emissions.

Q7. The article says that "at a provincial scale, only the data on cement production was available, with other indicators from the steel, chemical and other industries were missing." Does this refer to monthly data, or are indeed also no annual provincial data for these other sectors to be found? If annual provincial data for production in other industrial sectors are available, why are those not incorporated into the method to disaggregate the national industry emissions? Doing so would better constrain provincial estimates in this important sector compared to just using cement production.

**Response:** Thanks for your comments. Just as explained in the reply to the Q2, in this version of the manuscript, the provincial emission distribution in the industrial sector includes the data of 35 types of industrial products, covering nearly 90% of the emission in industrial sector.

*Q8. Just before the concluding section, there is a short section discussing uncertainty. No quantification of the uncertainty is presented here. Only a very small number of the sources of uncertainty are briefly noted, and Monte Carlo anslysis is mentioned, without any results being presented. This is far from sufficient.*

**Response:** Thanks for your comments. We have provided more detailed discussions of uncertainties in Q1.

Q9. Overall, my sense is that this paper takes the approach that if a method can be developed then the numbers should be shown, regardless of how heroic assumptions might be. If I had a method with numerous untested assumptions to estimate some number that allowed me to produce that number with 10 significant figures, and then used a further method with numerous untested assumptions to extend that to 15 significant figures, should I then report that number at 15 significant figures? The analogy here is to both temporal and spatial scales: while the methods developed allow the calculation of daily provincial emissions, this is equivalent to my example of calculating something to significant figures. Having a method is not the same as having a reasonable method. The authors started the article arguing that more accurate and more timely estimates were needed for provincial emissions, but the method only appears to address the latter (timeliness) and ignores the former (accuracy).

**Response:** As your comments pointed out, this study focus on the timeliness of current inventories, in the last manuscript, we use only cement and vehicle ownership as provincial proxies to disaggregate national daily data. And we also have a substantial light discussion on uncertainties and comparison of the other datasets, which makes us seem to overlook the accuracy of this dataset. In order to the accuracy of our method, we expanded the range of proxies (please see more detail in Q2, Q4) and quantified the uncertainties (more detail in Q1), and add the comparison of the other three datasets (Q12).

Q10. Much of the results section analyses patterns over the year, which does not require daily estimates, which is good. If the paper is rewritten to be an estimate of monthly provincelevel emissions, this reviewer would be more satisfied of its contribution.

**Response:** The impact of the short-term event (such as some public events, like the COVID-19, or some long holidays, like the Chinese New Year) on carbon emission can hardly be reflected by monthly data. Although our research on the impact of these short-term events can hardly reflect the provincial differences in day to day variation of emission, the provincial distribution method can be taken to approximately estimate the provincial emissions of different sectors during the COVID-19 event, holiday, weekend, or other periods.

Q11.Line 57: "However, these studies are based on provincial energy statistics". The authors need to state clearly why this statement warrants 'however', with the implication that this is a poor approach to estimating emissions.

**Response:** The original sentence on Line 57 has been changed into "These researches are mostly based on the provincial energy statistical data sourced from the yearbooks. There is at least one-year time lag between the data reflected situation and the real situation, leading to an increase in the uncertainty in establishing the emission reduction target and measuring the emission reduction effect.".

Q12. On line 296 the authors write that a result is consistent with three previous studies, but since all three used the same underlying dataset (CarbonMonitor for China), this is to be expected and the three studies cannot be used to support the result. Exploring consistency with other studies relies on use of independent datasets and/or methods.

**Response:** An additional comparison between this study and another two databases (MEIC and PKU) at the provincial level of China has been added in addition to the comparison between the research result and the CEADs in the old version. As the statistics of the different database was conducted in a different year, the comparison should be conducted on the basis of an assumption that "there was the same annual rate of variation in emission", which however was quite strict. As revealed in different databases, there is a very small inter-annual variation in the provincial contribution of annual emission (Table 7). In view of this, a comparison was made on the provincial contributions of this study and other datasets to reflect the difference in the provincial emission distribution in this study and other studies. We found that the emission contributions of Jilin Province and Heilongjiang Province estimated in this research are obviously underestimated (for almost twice) compared with that estimated in other researches. The emission contributions of other provinces (except Tibet) estimated in this research are consistent with those shown in other databases. On a monthly scale, our results were compared with the results of MEIC (Fig.4-7). The comparison results show that this research data is consistent with the MEIC data in terms of the variation trend of provincial monthly emission contribution (Fig.4). For power sector (Fig.5), the two databases are relatively approximate as can be seen from the trend of provincial monthly emission contribution; there is a large difference between Inner Mongolia and Liaoning with respect to the monthly contribution. For industrial sector (Fig.6), there is a difference in the monthly variation trend between Jiangsu, Guangdong, Guizhou, and Xinjiang, and a large difference in the emission contribution between Heilongjiang, Hubei, and Hebei. For transportation sector (Fig.7), no significant difference in provincial monthly emission is shown in the MEIC database. However, we found that more than 1% fluctuation is seen in the monthly contribution to emissions in Liaoning, Guangdong, Zhejiang, Shaanxi and Guizhou, and other regions.

Q13. It is very good to see that Tibet is included, for completeness, although obviously that's achieved by way of very approximate proxies since energy data are not available. I would like to see it mentioned that Tibet's emissions are the lowest of all provinces. However, I see that CEADS has Tibetan emissions in 2014, so this article is not the first to publish Tibet's emissions. Please rephrase.

**Response:** Thanks for your comments on the emission in Tibet. Indeed, there is data on the emission in Tibet as disclosed on the CEADs in 2014 and estimated on another two databases. However, in this research and the research of CEADs, the estimation of emission in Tibet was made on the basis of statistical data, while that estimation in MEIC and PKU was based on the sum of grid data. As discovered in the databases, the provincial emission contribution of Tibet is quite small (all 0%). Assuming that the inter-annual variation in emissions of Tibet is small, the emission of Tibet estimated in this research is close to the results shown in CEADs and MEIC, but more than 4 times different from the result shown in PKU. In addition, the emission of the different sectors in Tibet estimated in this research was also compared with that provided in the CEADs (Fig.7).

Q14. In the section "Trends in emissions from industry sector" the authors are effectively comparing the ratio of monthly provincial cement emissions to national total industrial emissions, because of the way provincial industrial emissions are calculated, and all trends discussed are of that ratio. Given this, do the authors believe that their interpretations in this section are useful? I do not.

**Response:** Thanks for your comments. Just as explained in the reply to the Q2, in this version of the manuscript, the provincial emission distribution in the industrial sector includes the data of 35 types of industrial products, covering nearly 90% of the emission in industrial sector.

Q15. The results presented in the section "Trends in emissions from ground transport" are entirely national, which is to be expected given the method uses no additional information on sub-annual emissions at the provincial level over what the national-level data already provide. This should be made clear to avoid readers misunderstanding.

**Response:** In the last manuscript, daily trends in provincial emissions from ground transport largely depended on national change. In the new manuscript, we have enlarged the provincial indicators to emissions from ground transport. We collected 12 provincial indicators to allocate national emissions, 3 indicators at monthly scale, 1 indicator at the seasonal scale, and 8 indicators at yearly scale. In these 8 yearly national indicators, two indicators, passenger traffic, and goods traffic are two consecutive years, and the others are for 2019 (Table 3). Although it still shows the provincial daily trends in CO2 emissions from ground transport sector because of only

3 monthly provincial indicators related to CO2 emissions from ground transport. But as we mentioned in Question 5, daily changes in carbon emissions from ground transport sector are more susceptible to the effects of holidays and Mondays, which may not be very heterogeneous at the provincial scale. According to Table 5, the impact of holidays and Mondays accounted for more than 60% of the daily change of CO2 emissions from ground transport.

Q16. In the analysis of the effects of holidays, results are presented in the text only for how the reduced emissions during holidays compare to the annual total, which is of minor interest. That the most significant holiday in 2019 reduced emissions by 0.33% isn't very interesting in itself. That this holiday was 7 days in duration and nevertheless reduced emissions by little more than one day's normal emissions (1/365=0.27%) is perhaps more interesting, and also how much these holidays reduce daily emissions compared to the normal for that time of year would be of interest. Also of interest would be some discussion of what sorts of activities (e.g. cement and steel factories) continue regardless of holidays, and which activities decline (e.g. road passenger transport?), citing any literature on the subject.

**Response:**

As you pointed out, the reductions of emissions during Spring Festival were more than one day's normal emissions, especially in 2020, the duration of Spring Festival coincided with the lockdown of COVID-19, the reduced emissions equaled around two days emissions (Table 8 and Table 9). Compared with the power sector and industry sector, emissions from ground transport dropped even more in the Spring Festival of 2020. For the long holiday (Mid Autumn and National Day, 2020, 8 days), the steelmakers maintain normal production during holidays, the manufacturing industry has holidays with an average of 1-3 days, and no holidays for construction and infrastructure industries (https://news.metal.com/newscontent/101278794/A-summary-of-production-situation -in-the-steel-industry-chain-before-the-holiday/).

Q17. In the light of the very large uncertainty of the numbers, it is not appropriate to present results such as "91350.96 thousand tons". Please reduce the precision. Further, I presume these are tonnes (called 'metric tons' in the US) rather than tons, but if you do mean tons then you should make that clear.

**Response:** Thanks for your comments. More improvements about uncertainty in Question 1. And we have changed all tonnes to metric tons.

Q18. Line 414: "changed little in those two years." Which two years? The two years of this analysis, i.e. 2019 and 2020? If so, then your statement is that because there was little change in proportions between those two years, you assume that there was little difference between 2018 and 2019. This could be written more clearly rather than leaving the reader to interpolate. Further, CEADS has provincial

emissions from 1997 and the authors should use the information therein to support their assumption that provincial shares change little from year to year.

**Response:** We have reorganized the section on comparison of the other datasets, and this previous description has been removed. For more details on the conclusions of the model comparisons, please see the answer to question 12.

Q19. I think the authors should also provide the basic information for why this comparison is being made: Is CEADS the only other dataset to provide provincial emissions for China? Is it widely used? Is it considered a standard by which to compare?

**Response:** We use Table 2 to demonstrate the basic information of the other three datasets, including the item of the domain, temporal coverage and resolution, spatial resolution, data source, references, emission sectors, and the method of emission calculation.

Q20. The final sentence of the conclusions states "more work is still required in order to improve the provincial daily CO2 emission estimates from the lower emitting sectors, such as the residential, aviation and shipping sectors". I'm greatly concerned that the authors do not think that any further work is required to improve the sectors that are covered by the paper, given their very approximate nature.

**Response:** Thanks for your comments. In subsequent research, there are the following problems to be further improved: First, regarding the 9 industrial subsectors, it was assumed that different products had the same emission contribution to this sector. However, the different product has different emission factor. In subsequent research, it is needed to make clear the contributions of the emission factors to the subsectors to reduce the uncertainty caused by this assumption. Second, in the emission estimation for transportation sector, TomTom database was used to estimate the traffic flow in China. However, the TomTom database intrinsically only contains the data of 22 cities in China, so that the estimation based on this database is biased. We hope to find an alternative containing the data of more provinces in China in place of TomTom database in subsequent research. Third, among the provincial distribution indexes of transportation sector, the monthly data was based on the provincial productions of gasoline, diesel, and automobiles. But, the emission of transportation sector may be more relevant to the provincial consumption of gasoline, diesel, and automobiles.

**Figures**